# Time-dependent assessment of stimulus-evoked regional dopamine release

Rachel N. Lippert [1], Anna Lena Cremer[1], Sharmili Edwin Thanarajah[1,2], Clio Korn[3,4], Thomas Jahans-Price[4], Lauren M. Burgeno[4], Marc Tittgemeyer[1,5,6], Jens C. Brüning[1,5,7], Mark E. Walton[4,8] & Heiko Backes[1]

To date, the spatiotemporal release of specific neurotransmitters at physiological levels in the human brain cannot be detected. Here, we present a method that relates minute-by-minute fluctuations of the positron emission tomography (PET) radioligand [11C]raclopride directly to subsecond dopamine release events. We show theoretically that synaptic dopamine release induces low frequency temporal variations of extrasynaptic extracellular dopamine levels, at time scales of one minute, that can evoke detectable temporal variations in the [11C] raclopride signal. Hence, dopaminergic activity can be monitored via temporal fluctuations in the [11C]raclopride PET signal. We validate this theory using fast-scan cyclic voltammetry and [11C]raclopride PET in mice during chemogenetic activation of dopaminergic neurons. We then apply the method to data from human subjects given a palatable milkshake and discover immediate and—for the first time—delayed food-induced dopamine release. This method enables time-dependent regional monitoring of stimulus-evoked dopamine release at physiological levels.

[1] Max Planck Institute for Metabolism Research, Gleueler Str. 50, 50931 Cologne, Germany. [2] Department of Neurology, University Hospital of Cologne, Kerpener Str. 62, 50937 Cologne, Germany. [3] Department of Psychiatry, University of Oxford, Warneford Hospital, Oxford OX3 7JX, UK. [4] Department of Experimental Psychology, University of Oxford, Tinsley Building, Mansfield Road, Oxford OX1 3SR, UK. [5] Cologne Cluster of Excellence in Cellular Stress and Aging-Associated Disease (CECAD), Joseph-Stelzmann-Str. 26, 50931 Cologne, Germany. [6] Modern Diet and Physiology Research Center, 290 Congress Avenue, New Haven, CT 06519, USA. [7] Center for Endocrinology, Diabetes and Preventive Medicine (CEPD), University Hospital of Cologne, Kerpener Str. 62, 50937 Cologne, Germany. [8] Wellcome Centre for Integrative Neuroimaging, Department of Experimental Psychology, University of Oxford, Tinsley Building, Mansfield Road, Oxford OX1 3SR, UK. These authors contributed equally: Rachel N. Lippert, Anna Lena Cremer. Correspondence and requests for materials should be addressed to H.B. (email: backes@sf.mpg.de)

The neurotransmitter dopamine (DA) plays a key role in the control of motor function, motivation, food intake, and reward[1–3]. Malfunctions in the dopaminergic system cause severe symptoms and debilitating diseases (e.g., Parkinson's disease)[4,5]. The dopaminergic system is one of the most extensively studied neurotransmitter systems, nevertheless, it is still far from being fully understood. One reason originates from the basic properties of neurotransmitter signaling: in response to perception of macroscopic stimuli, such as sensory cues, molecules are released into synapses, only 20–30 nm broad, bind within milliseconds to intrasynaptic receptors, diffuse into extracellular space and bind to extrasynaptic receptors, trigger secondary processes, and eventually cause changes in macroscopic behavior. The whole cascade of signal transduction from stimulus to behavior includes a vast range of temporal and spatial scales but the available methods for the analysis of the processes can only address singular aspects of these complex events.

Using positron emission tomography (PET) and the radiotracer [11C]raclopride, we introduce here a method for the in vivo assessment of time-dependent regional dopamine release that makes use of the relation between different time scales in the dopaminergic system and that is—as we also demonstrate here—readily applicable to humans. [11C]raclopride is a well-known antagonist for dopamine type 2 receptors (D2Rs) and to lesser extent dopamine type 3 receptors[6]. Due to its relatively low binding affinity, it has, to date, been predominantly used for the quantitative steady-state assessment of available D2R binding sites in the striatum, the brain region with the highest density of D2Rs. [11C]raclopride competes with endogenous DA for binding to D2Rs and therefore binding events depend on extracellular concentrations of DA. The release of DA, and subsequent binding to the D2R, reduces the amount of free D2Rs available for [11C]raclopride interaction and thereby reduces the amount of bound [11C]raclopride[7]. Based on this principle, several models have been developed that relate task-induced or pharmacological reductions of [11C]raclopride binding to DA release[7–17]. However, given the relatively slow kinetic rate constants of [11C]raclopride, these methods require substantial, pharmacologically enhanced, and long-lasting DA release events. Changes in [11C]raclopride binding are thought to be insensitive to high-frequency transient DA variations.

Here, we propose and validate a novel approach for the analysis of [11C]raclopride data based on theoretically derived predictions of the spatial and temporal consequences of DA release events. We first demonstrate, with the help of a simple model based on fundamental principles, how low-frequency variations of extracellular DA concentrations can be directly linked to synaptic DA release. We then extrapolate how temporal variations of extracellular DA concentrations induce temporal variations of [11C]raclopride binding and derive a parameter that captures temporal variations of the [11C]raclopride signal and can therefore be used as an indirect measure for regional DA dynamics.

To support our theory, we performed [11C]raclopride PET in mice that carry the chemogenetically activatable modified muscarinic receptor (hM3D$_{Gq}$) exclusively in DA neurons. We then compared these [11C]raclopride PET data with sub-second recordings of DA concentrations in the ventral striatum measured in situ using fast-scan cyclic voltammetry (FSCV). We demonstrate that (i) chemogenetic activation of DA neurons in mice increases the rate of spontaneous DA transients detected by FSCV and that (ii) the number of transients is significantly correlated with the logarithm of the power in the frequency band of 0.5 Hz, as calculated by wavelet transform of continuously recorded FSCV data. We further show that (iii) the logarithm of high-frequency power (~0.5 Hz) and transient rates correlate

significantly with the logarithm of low-frequency power (~0.01 Hz) and that (iv) these low-frequency variations of extracellular DA concentrations cause detectable variations in the [11C]raclopride PET signal. With the mouse data supporting our theory we conclude that temporal variations of the [11C]raclopride PET signal can be used as a measure of dopaminergic activity.

Finally, we applied our method to healthy human volunteers who received a palatable milkshake during [11C]raclopride PET data acquisition and identified acute and delayed stimulus-related DA release in multiple striatal and extrastriatal brain regions.

## Results

**Low-frequency variations of extracellular DA.** During a synaptic release event, DA concentrations are elevated $10^4$-fold within the synaptic cleft for ~1 millisecond[18] (Fig. 1a). Synapses are not leak-proof and due to the high concentration gradient at the border of the synapses part of the released DA diffuses into extracellular space[19]. These changes in extracellular DA concentrations generate the signals detected by FSCV that show stimulus-related transient 10–100-fold increases of extracellular DA levels that last for ~2 s (Fig. 1b). Indeed, shape, magnitude, and duration of FSCV transients can be precisely modeled as integrated DA from synapses surrounding the probe leaking into extracellular space. The extracellular DA concentration $D_e$ is then given by[18]:

$$D_e(r,t) = \sum_{i,j(t>t_{i,j})} \frac{l_s V_s C_s}{\alpha \left(4\pi D\left(t - t_{i,j}\right)\right)^{3/2}} exp\left(\frac{-(r - r_i)^2}{4D\left(t - t_{i,j}\right)}\right) exp\left(-k\left(t - t_{i,j}\right)\right)$$

(1)

$V_s$ is the Volume of the synapse, $C_s$ the DA concentration within the synapse, $D$ the apparent diffusion coefficient, $\alpha$ the fraction of extracellular volume, and $l_s$ the fraction of synaptic leakage or spillover (typical values for the striatum: $V_s = 0.02$ μm$^3$, $C_s = 0.8$ mmol/L, $D = 7.63e{-}6$ cm$^2$/s, $\alpha = 0.21$, $l_s = 0.01$)[18,19]. Note that Equation (1) strictly applies to dopamine diffusion in the extracellular space outside the synapse. Within the synapse the narrow space confined by pre- and postsynaptic membranes and the high density of transporters and receptors would require a mathematical description different from Equation (1). However, Equation (1) successfully describes the dopamine concentrations in the extracellular space observed after phasic release. The fraction, $l_s$, that was able to diffuse out of the synapse, and not the total amount of DA released, should then be inserted into Equation (1). The model includes DA diffusion from the synapses at locations, $r_i$, and removal from extracellular space by DA transporters (DATs) with an effective removal rate constant of $k$ reflecting the local density of DATs. Reported values range from $k = 0.01$ s$^{-1}$ to 20 s$^{-1}$ corresponding to effective lifetimes of 100 to 0.05 s[18,20]. The summation is over all synapses, $i$, and release times, $j$, with $t_{i,j}$ being the times of DA release of a synapse at location $r_i$. See Fig. 2 for DA concentration at the location of an FSCV probe as a function of time after synaptic DA release calculated from Equation 1 assuming that the probe receives input from a fast domain with a high density of DATs ($k = 2$ s$^{-1}$) and a slow domain with a low density of DATs ($k = 0.02$ s$^{-1}$). The absolute rate of removal in each domain is given by the product of removal rate constant and extracellular DA concentration ($-kD_e$). This means that, depending on the local density of DATs, net changes of extracellular DA concentrations occur at time scales of minutes to seconds and the amplitude of these changes ($dD_e(r,t)/dt$) is proportional to extracellular DA concentration ($D_e(r,t)$) which, by its origin, is directly proportional to synaptic DA release. Thus, the amplitude of low-frequency

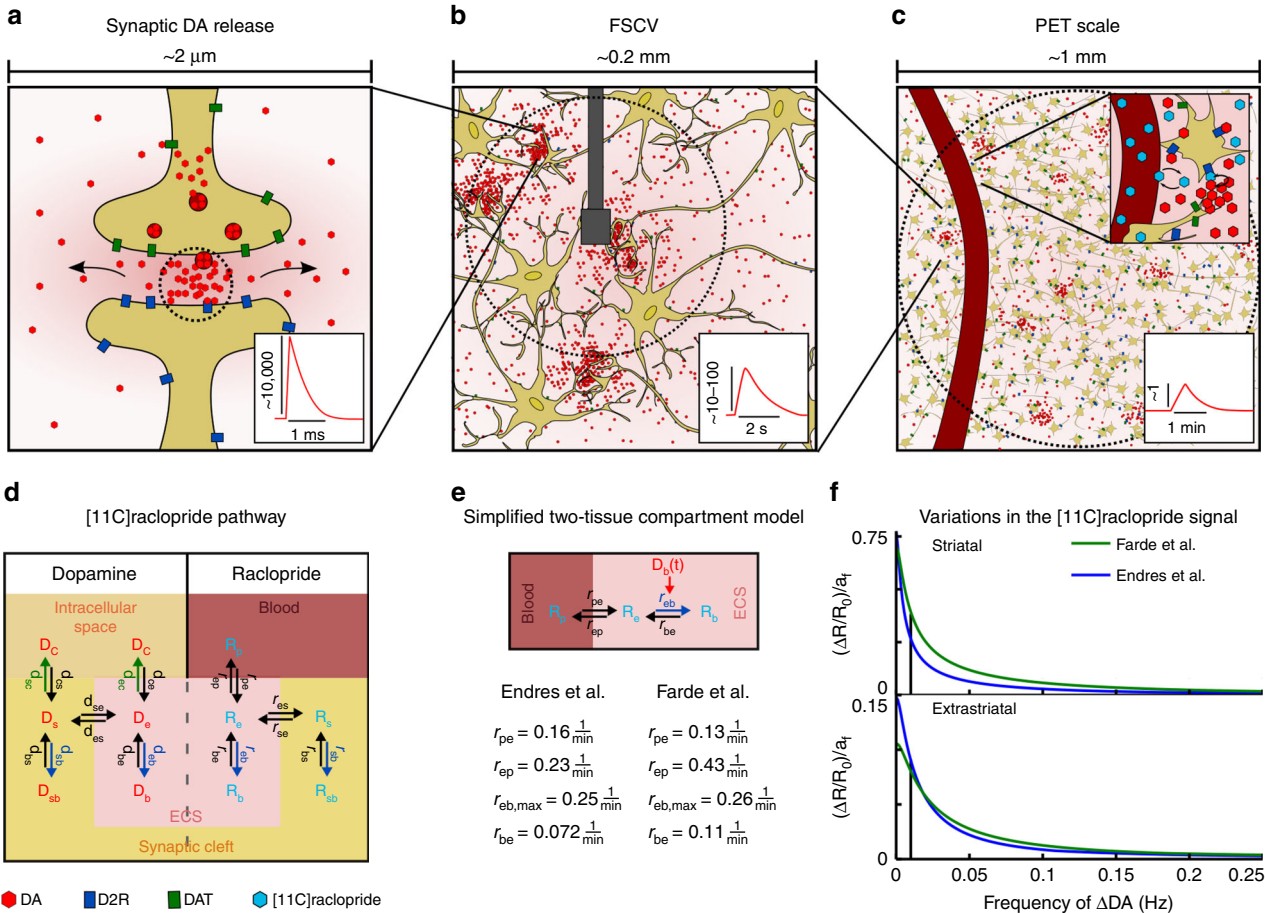

**Fig. 1** DA release causes spatiotemporal variations of DA concentrations on multiple scales. **a** DA release increases synaptic DA concentrations by a factor of ~10,000 for ~1 ms. **b** Diffusion-driven outflow of DA from synapses increases DA concentration 10–100-fold in the close vicinity of synapses for ~2 s. These transients can be directly measured using FSCV. **c** DA is cleared from extracellular space by dopamine transporters (DAT). The time scale for clearance depends on the local DAT density. In a volume of 1 μL part of the DA from synaptic release diffuses into regions with low DAT density and accordingly slow clearance rates of ~1 min. In the extracellular space DA competes with [11C]raclopride for binding to D2 receptors. Unbound [11C] raclopride concentration is in equilibrium with [11C]raclopride concentration in the blood (vessel shown in dark red). **d** Synaptic and extracellular tissue compartments are shared by DA and [11C]raclopride, DA is supplied from and cleared into axons (intracellular space), [11C]raclopride from the blood. DA compartments are labeled with D, transfer rate constants with d, [11C]raclopride compartments with R and transfer rate constants with r. Subscripts are c: intracellular space, p: plasma, s: synapse, e: extracellular (extrasynaptic) space, b: bound to D2 in ECS, sb: bound to D2 in synapse. **e** Effectively the [11C] raclopride PET signal is determined by the ECS. Temporal variations of D2-bound extrasynaptic DA modulate the [11C]raclopride binding constant $r_{eb}$. Reported kinetic rate constants according to Endres et al.[21] and Farde et al.[22]. **f** Periodic changes of $D_b$ cause frequency-dependent periodic variations of the [11C]raclopride PET signal that are proportional to the amplitude $a_f$ of $D_b$ variations using kinetic rate constants from Farde et al. (green line) and from Endres et al. (blue line). $a_f$ can have values between 0 and 1, $a_f = 1$ means that DA occupies 100% of the D2 receptors. Only variations at low frequency are detectable by PET. Extrastriatal variations were calculated assuming a D2R density of 5% of the striatal density

variations of extracellular DA concentrations is directly proportional to (and is therefore a potential measure of) synaptic DA activity (Fig. 1a–c, Fig. 2).

**Variations of extracellular DA cause PET signal variations.** After intravenous injection, [11C]raclopride is transported to the brain, where blood and brain concentrations in the extracellular fluid equilibrate. Within the brain, [11C]raclopride diffuses through extracellular space which effectively consists of two compartments: the extracellular extrasynaptic compartment covering the fractional volume $v_e$ (referred to as ECS) and the synaptic compartment covering the fractional volume $v_s$. Within each compartment [11C]raclopride binds to D2R according to its binding affinity which leads to the reaction pathway illustrated in Fig. 1d.

Endogenous DA, released into synapses or directly into the ECS, partially occupies the same compartments as [11C]

raclopride. Diffusion of DA between synapses and ECS causes variations of DA concentration in the ECS even in the absence of direct DA release into the ECS. DATs transport DA from ECS and synapses back into the cells ($d_{ec}$ and $d_{sc}$ in Fig. 1d). In the following model, we only take into account DA binding to D2Rs omitting other receptor types due to the specific competition with [11C]raclopride binding. As illustrated in Fig. 1d, DA and [11C] raclopride both occupy ECS and synapses and compete for binding to D2R within these compartments.

The effective binding rates of [11C]raclopride ($r_{eb}$) and DA ($d_{eb}$) to D2R in the ECS depend on the amount of unbound D2R:

$$r_{eb} = r_{eb,max}\left(1 - \frac{D_b}{T_e} - \frac{R_b}{T_e}\right) \quad (2)$$

$$d_{eb} = d_{eb,max}\left(1 - \frac{D_b}{T_e} - \frac{R_b}{T_e}\right) \quad (3)$$

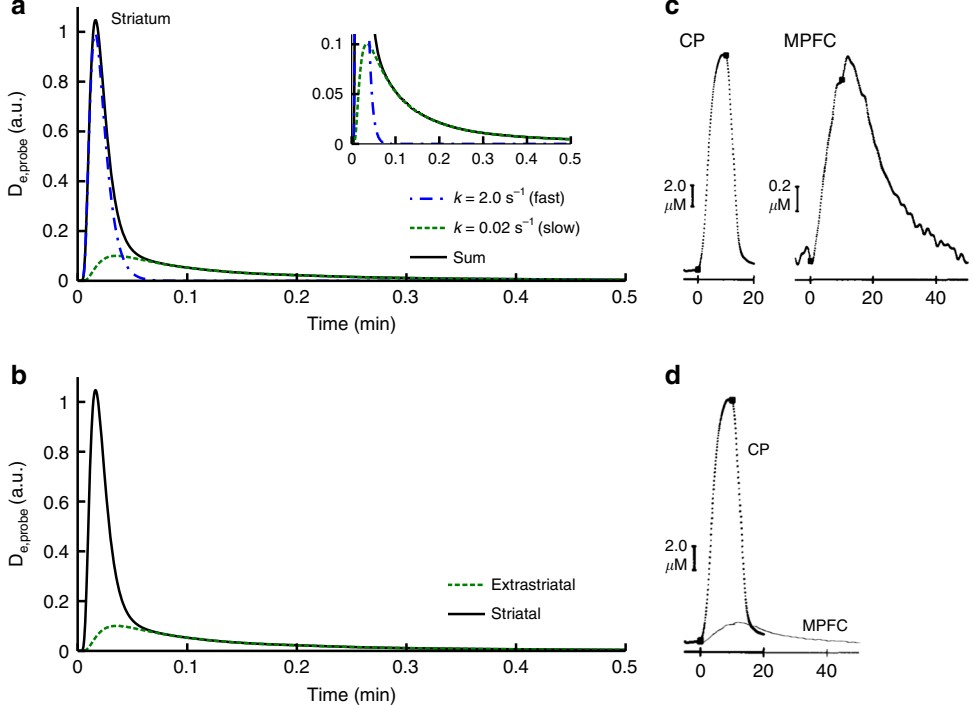

**Fig. 2** Extracellular DA after synaptic release. **a** Calculated extracellular DA concentrations at the location of an FSCV electrode (black, solid) assuming that the signal contains contributions from a fast domain (blue, dash-dot) with high density of DA transporters (DATs) and a slow domain (green, dashed) with low density of DATs (Equation 1). DA is removed from the fast domain at a rate of s⁻¹ and from the slow domain at a rate of min⁻¹. Although the overall FSCV signal in the striatum is dominated by the signal from the fast domain, the slow domain determines variations of extracellular DA concentration on a minute-by-minute scale. The amplitudes of both, the fast and the slow domain, depend on synaptic DA release. **b** The peak amplitude of dopamine transients in extrastriatal regions is lower due to the lower spatial density of DA synapses (green, dashed). All of the released DA is removed at minute time scale due to the lower density of DATs in extrastriatal regions in comparison to the striatum. Consequently the amplitude of minute-by-minute changes in extracellular DA concentration is the same in extrastriatal and striatal regions (black, solid). **c** Simultaneous FSCV recordings from the striatum (CP) and the medial prefrontal cortex (MPFC) in rats after electrical stimulation of the VTA by Garris et al. support this theory[42]. In order to evoke a detectable signal in the MPFC the VTA was stimulated at superphysiological amplitudes. **d** Combination of the CP and MPFC traces in a single graph indicates that the tail representing slowly removed DA in the CP is of the same magnitude as the extrastriatal signal

$T_e$ is the total (bound plus unbound) amount of D2Rs located on extrasynaptic cellular membranes and thereby accessible by DA and [11C]raclopride in the ECS. $D_b$ and $R_b$ are the concentrations of bound DA and bound [11C]raclopride, respectively. $r_{eb,max}$ and $d_{eb,max}$ are the maximum binding rates of [11C]raclopride and DA when the total amount of D2R is available for binding. The total [11C]raclopride concentration, $R$, in a certain volume is the sum of all compartments:

$$R = v_P R_P + v_e(R_e + R_b) + v_s(R_s + R_{sb}) \quad (4)$$

$R_P$ is the concentration of [11C]raclopride in plasma, $R_e$ the free concentration in extracellular extrasynaptic space, $R_s$ the free concentration in the synapses, and $R_{sb}$, the concentration of intrasynaptic bound [11C]raclopride. Since [11C]raclopride does not diffuse into blood cells, the contribution of [11C]raclopride in blood to the volume signal, $R$, is given by the product of plasma volume fraction $v_P$ and plasma concentration $R_P$. Given that the free concentrations in tissue are equilibrated, i.e., $R_e = R_s$ and that the amount of bound [11C]raclopride ($R_b$) is in the order of magnitude of free [11C]raclopride, the contributions of the synaptic [11C]raclopride to the volume signal can be neglected due to the small volume of synapses relative to ECS volume[19]. This leads to a simple two-tissue compartment model illustrated in Fig. 1e. Time-dependent changes of [11C]raclopride

concentrations in the free and bound ECS compartment are given by:

$$\frac{dR_e}{dt} = r_{pe}R_p - \left( r_{ep} + r_{eb,max}\left(1 - \frac{D_b}{T_e} - \frac{R_b}{T_e}\right)\right) R_e + r_{be}R_b \quad (5)$$

$$\frac{dR_b}{dt} = r_{eb,max}\left(1 - \frac{D_b}{T_e} - \frac{R_b}{T_e}\right)R_e - r_{be}R_b \quad (6)$$

[Note that for DA, in contrast to [11C]raclopride, the contribution of the synaptic volume would not be negligible because DA concentrations within synapses reach levels 3–4 orders of magnitude higher than in the extracellular space.] Since the [11C]raclopride signal effectively originates from the ECS, it is influenced only by variations of DA binding in the ECS and not in the synapse. For further simplification, we assume that the fraction of D2Rs blocked by [11C]raclopride is negligible in comparison to the total amount of D2Rs. If the amount of D2R-bound DA in the ECS varies in time ($D_b(t)$), the number of D2Rs available to [11C]raclopride changes and the binding parameter of [11C]raclopride ($r_{eb}$) varies in time by the factor $f_{DA}(t) = (1-D_b(t)/T_e)$. The effective equations that describe [11C]raclopride kinetics in tissue taking into account DA dynamics are then given

by:

$$\frac{dR_e}{dt} = r_{pe}R_p - \left( r_{ep} + r_{eb,max}f_{DA}(t) \right)R_e + r_{be}R_b \qquad (7)$$

$$\frac{dR_b}{dt} = r_{eb,max}f_{DA}(t)R_e - r_{be}R_b \qquad (8)$$

If we further assume the contribution of [11C]raclopride in the blood is negligible due to the small fractional volume of this compartment (~3%), the total [11C]raclopride signal in a tissue volume is $R(t) = R_e(t) + R_b(t)$. In order to estimate the impact of temporal variations of DA on the detectable [11C]raclopride signal we decompose $f_{DA}(t)$ as a sum of harmonic oscillations:

$$f_{DA}(t) = \frac{1}{2}\left( 1 + \sum_f a_f sin(2\pi ft) \right) \qquad (9)$$

We solved Equations 7 and 8 numerically for different frequencies, $f$, and different amplitudes, $a_f$, and found the relation between variations of $R(t)$ and the frequency, $f$, of temporal variations of DA shown in Fig. 1f. This relation informs us that, for example, variations of $D_b/T_e$ with a frequency of 0.25 Hz and an amplitude of 1 (number of D2Rs bound to DA fluctuates from 0% to 100%) will cause negligible relative variations in $\Delta R(t)/R_0$. $D_b/T_e$ variations with the same amplitude but a frequency of 0.01 Hz induce in the striatum variations in $\Delta R(t)/R_0$ of 0.25 if we adopt the [11C]raclopride kinetic parameters from Endres et al. or 0.38 with the parameters taken from Farde et al.[21,22]. [11C] raclopride variations scale linearly with the amplitude $a_f$ of DA variations, i.e., the term $\Delta R(t)/R_0/a_f$ shown in Fig. 1f only depends on the frequency, $f$, but is independent of the amplitude, $a_f$, of DA variations. At an amplitude of 0.5 (number of D2R bound to DA changes periodically from 25% to 75%) $\Delta R(t)/R_0$ are 0.125 (Endres et al.) or 0.19 (Farde et al.) at 0.01 Hz. The density of D2Rs in extrastriatal regions is ~2%-8% of the striatal density[23]. Accordingly, extrastriatal DA variations at an amplitude of 1 induce $\Delta R(t)/R_0$ at 0.01 Hz of ~0.08 (Fig. 1f).

**Assessment of regional temporal [11C]raclopride variations.** Temporal variation of the [11C]raclopride PET signal at time $t_n$ and location $i,j,k$ can be calculated as:

$$rDA \equiv \frac{\Delta R_{ijk}}{R_{0,ijk}}(t_n) = \frac{1}{R_{0,ijk}}\sqrt{\frac{1}{N_{sum}}\sum_{u=i-d_i}^{i+d_i}\sum_{v=j-d_j}^{j+d_j}\sum_{w=k-d_k}^{k+d_k}(R_{uvw}(t_n)-R_{uvw}(t_{n-1}))^2}$$

$$(10)$$

$$R_{0,ijk} = \frac{1}{N-ns}\frac{1}{N_{sum}}\sum_{n=ns}^{N}\sum_{u=i-d_i}^{i+d_i}\sum_{v=j-d_j}^{j+d_j}\sum_{w=k-d_k}^{k+d_k}R_{uvw}(t_n) \qquad (11)$$

The square root term is a measure of $\Delta R(t_n)$, the average change of the [11C]raclopride signal from time $t_{n-1}$ to time $t_n$ in the box centered at $i,j,k$ ($N_{sum}$, the number of voxels in the box is $(2d_i+1)(2d_j+1)(2d_k+1)$). $R_0$ is calculated in Equation 11 as the average signal in the box after the quasi steady state has been reached at $t_{ns} = 20$ min after bolus injection. See Methods section for further details.

In summary, by utilizing the theoretical considerations referenced above, we determined that [11C]raclopride effectively models dynamic variations of DA bound to D2Rs in the ECS at frequencies below 0.05 Hz. The relative amplitude of regional [11C]raclopride variations ($\Delta R(t)/R_0$) is directly proportional to the amplitude, $a_f$, of variations of DA bound to D2R in the ECS.

$\Delta R(t)/R_0$ can be calculated directly from dynamic PET data as a measure of DA release rates.

**Dopaminergic activity in the ventral striatum of mice.** In the previous sections, we demonstrated theoretically that synaptic DA activity is related to low frequency (~0.01 Hz) variations of extracellular DA which cause detectable temporal variations in the [11C]raclopride PET signal. Here we substantiate the theoretical considerations with data from electrically stimulated wild-type mice and a chemogenetic mouse model, in which dopaminergic cells express a modified muscarinic receptor (hM3D$_{Gq}$DAT) activated by injection of an exogenous synthetic compound, clozapine-n-oxide (CNO)[24]. After injection of CNO hM3D$_{Gq}$DAT mice displayed behavioral changes indicative of dopaminergic activation (Fig. 3; Supplementary Notes).

FSCV data were acquired continuously at a rate of 10 Hz in wild-type mice, where transient DA release was induced by electrical stimulation of the VTA, and in chemogenetically activated (hM3D$_{Gq}$DAT+CNO) and non-activated mice (hM3D$_{Gq}$DAT+Saline). With these data we show that (i) chemogenetic activation of DA neurons increases the rate, but not the magnitude, of spontaneous DA transients measured in the ventral striatum, that (ii) the rate of transients—electrically or chemogenetically induced—is correlated with high frequency (~0.5 Hz) variations in DA levels extracted from the continuous FSCV data, and that (iii) transient rates and high frequency variations are correlated with low frequency variations (~0.01 Hz). We further show that (iv) chemogenetic activation of DA neurons in the hM3D$_{Gq}$DAT mouse model increases temporal variations in [11C]raclopride PET signal.

**i. Activation of DA neurons increases rate of DA transients.** We first verified the electrode placement by examining changes in DA levels in the nucleus accumbens evoked by electrical stimulation of the ventral tegmental area (VTA). There were no differences in stimulated DA in the hM3D$_{Gq}$DAT+CNO or Saline-treated animals prior to drug administration (main effect of group: $F(1,4) = 1.33$, $p = 0.31$; group x treatment: $F(4,16) = 1.68$, $p = 0.20$, Two-way ANOVA; note that one mouse of the hM3D$_{Gq}$DAT CNO group was excluded from this analysis due to a slightly different stimulation protocol). We then determined the spontaneous dopamine transient rate in continuous FSCV data recorded before and after CNO application by correlating the recorded cyclic voltammograms with an evoked dopamine "template" (Fig. 4a; see Methods section). Since transients are directly linked to synaptic DA release, the number of transients per time is a measure of dopaminergic activity. The rate of transients was counted in one-minute intervals of FSCV data of CNO- and Saline-treated hM3D$_{Gq}$DAT mice. Despite minor differences in spontaneous activity in the first 5 min of baseline recording, the results show a significant increase of the DA transient rate following CNO-treatment that was not present in the Saline-treated mice (main effect of treatment: $F(1,5)_{1st\ hour} = 14.31$, $p = 0.0129$, $F(1,5)_{2nd\ hour} = 6.99$, $p = 0.0458$) (Fig. 4c). Nonetheless, there was no consistent change in the size of the release events in either group over the recording session comparing the pre-drug period with the 1st or 2nd hour after drug administration (period x group interaction: $F(2,8) = 0.946$, $p = 0.428$, Two-way ANOVA; note one saline-treated mouse had no transients in the pre-drug period so was excluded from this analysis), although the transient size recorded in the CNO-treated group was on average marginally higher throughout recording (main effect of group: $F(1,4) = 12.34$, $p = 0.025$) (Fig. 4b). Therefore, while CNO administration increased the transient rate

or probability, it had little effect on the amplitude of each measured release event.

**ii. DA transient rate correlates with DA variations at 0.5 Hz.** Next, to establish a link between high and low frequency variations of DA concentrations in the ECS and the rate of transients,

we extracted a continuous estimate of DA levels from the FSCV signal (Fig. 4a; see Methods section) and decomposed this continuous signal into contributions from different frequencies by performing a wavelet transform[25]. Figure 5 shows the trace of FSCV data continuously recorded in the ventral striatum of a single mouse (Fig. 5a) and the wavelet power spectrum calculated

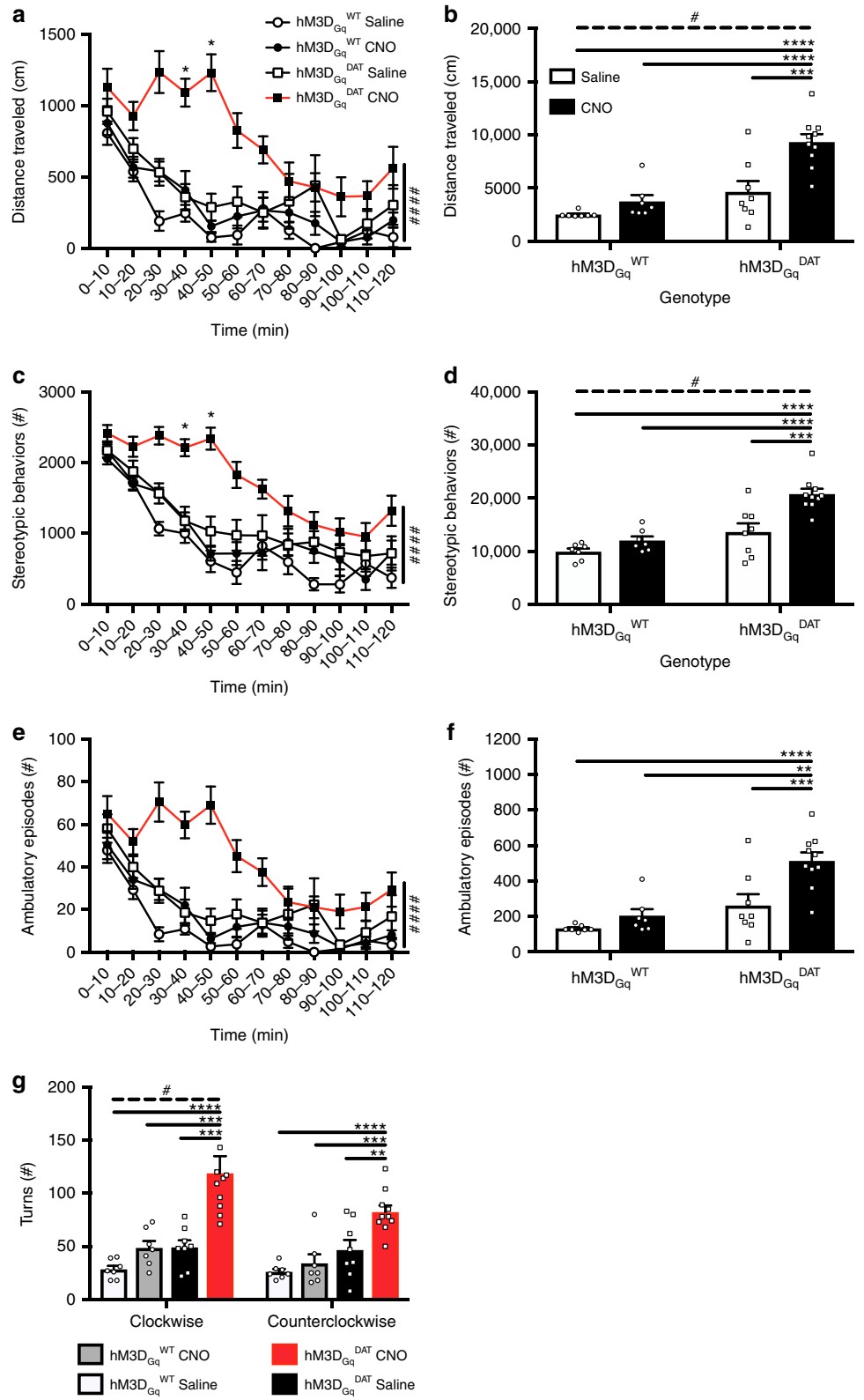

**Fig. 3** Locomotor activity in the open field in chemogenetically stimulated mice. **a**, **b** Distance traveled, **c**, **d** stereotypic behaviors and **e**, **f** ambulatory episodes over 120 min (**a**, **c**, **e**) and in total (**b**, **d**, **f**) in hM3D$_{Gq}$$^{DAT}$ and hM3D$_{Gq}$$^{WT}$ mice stimulated with Saline (10 μL/g BW) or CNO (0.3 mg/kg) at $t = 0'$ (open circle = hM3D$_{Gq}$$^{WT}$+Saline ($n = 7$), closed circle = hM3D$_{Gq}$$^{WT}$+CNO ($n = 7$), open square = hM3D$_{Gq}$$^{DAT}$+Saline ($n = 8$), closed square = hM3D$_{Gq}$$^{DAT}$+CNO ($n = 10$)). **g** Rotational behavior in the clockwise or counterclockwise direction throughout the duration of the open field test. Over time data was analyzed via Three-way ANOVA, with a Two-way ANOVA post-hoc analysis at each time point corrected for false discovery rate, * = $p < 0.05$ FDR (5%) corrected, #### = $p < 0.001$ for overall genotype x treatment interaction (**a**, **c**, **e**). Summation data were analyzed via Two-way ANOVA. Treatment and genotype effects were further compared with post-hoc pairwise comparisons via Tukey's multiple comparison test and Bonferroni corrected. All data are represented as mean±SEM. **** = $p < 0.001$, *** = $p < 0.005$, ** = $p < 0.01$, * = $p < 0.05$, # = $p < 0.05$ for overall genotype x treatment interaction (**b**, **d**)

from this trace using a Gaussian mother function of order 3 (Fig. 5b, c).

The typical duration of DA transients in FSCV data is ~2 s (Fig. 4b) causing an expected increase in the power spectrum at a frequency of 0.5 Hz. We tested the relation of transient rates and 0.5 Hz wavelet power in wild-type mice where definite rates of electrically stimulated DA transients were induced. Stimulations were performed at rates of ten per minute or five per minute for one-minute intervals followed by a four-minute (wt-mouse 1) or a nine-minute interval (wt-mouse 2, 3, and 4) without stimulation. The logarithm of the 0.5 Hz wavelet power in each one-minute interval was significantly correlated with the number of transients induced during this interval (Pearson product moment correlation: $r = 0.89$, $p < 10^{-15}$) (Fig. 6).

In agreement with electrically induced DA release in wild-type mice, transients counted in one-minute intervals in chemogenetically activated mice were significantly correlated with the logarithm of the wavelet power of DA variations at frequencies of ~0.5 Hz on group level but also in individual mice (Pearson product moment correlation: $r_{CNO} = 0.64$, $p_{CNO} < 10^{-16}$, $df_{CNO} = 296$, $t_{CNO} = 14.26$, $r_{Saline} = 0.63$, $p_{Saline} < 10^{-16}$, $df_{Saline} = 215$, $t_{Saline} = 11.75$, $r_{combined} = 0.73$, $p_{combined} < 10^{-16}$, $df_{combined} = 513$, $t_{combined} = 24.44$) (Fig. 7, Supplementary Figure 1 and 2). This argues that the power at 0.5 Hz in FSCV data, which can be calculated following a well-defined standard procedure and does not require any thresholds or assumptions on the shape of transients, can be used as a measure of the transient rate. Consequently, activation of DA neurons in mice increases not only the transient rate, but also the wavelet power at 0.5 Hz.

### iii. Transient rates and high and low-frequency variations.
Decomposition of continuous FSCV data into contributions from different frequencies allows for correlation analysis between high-frequency and low-frequency contributions, which we tested between the logarithm of the FSCV wavelet power at 0.5 Hz (referred to as "high") and 0.01 Hz (referred to as "low"). In electrically stimulated mice we found a significant correlation between the logarithm of high and low frequency power ($r = 0.55$, $p = 0.00012$; Fig. 6e) and also between transient rates and the logarithm of low frequency power (transient rates vs. log(0.01 Hz power), Pearson product moment correlation: $r = 0.44$, $p = 0.0031$).

Furthermore, with chemogenetically activated mice, transient rates and the logarithm of high and low frequency wavelet power correlated significantly within each group and in the combined data (Pearson product moment correlation between logarithmic power at 0.5 Hz and 0.01 Hz: $r_{CNO} = 0.13$, $p_{CNO} = 0.030$, $df_{CNO} = 296$, $t_{CNO} = 2.19$, $r_{Saline} = 0.14$, $p_{Saline} = 0.042$, $df_{Saline} = 215$, $t_{Saline} = 2.05$, $r_{combined} = 0.34$, $p_{combined} < 10^{-14}$, $df_{combined} = 513$, $t_{combined} = 8.17$; correlation between transient rates and logarithmic power at 0.01 Hz: $r_{CNO} = 0.15$, $p_{CNO} = 0.008$, $df_{CNO} = 296$,

$t_{CNO} = 2.69$, $r_{Saline} = 0.16$, $p_{Saline} = 0.016$, $df_{Saline} = 215$, $t_{Saline} = 2.43$, $r_{combined} = 0.32$, $p_{combined} < 10^{-12}$, $df_{combined} = 513$, $t_{combined} = 7.63$) (Fig. 5d, Fig. 8a–d). Consistently, activation of DA neurons not only increased temporal fluctuations on a time scale of one second (as measured by the high-frequency wavelet power), but also systematically induced fluctuations at a minute time scale (as measured by the low-frequency wavelet power).

### iv. DA release increases temporal variations in PET signal.
[11C]raclopride PET emission data were acquired in the same mouse model using the same protocol with respect to hM3D$_{Gq}$$^{DAT}$ activation (Methods section). To rapidly reach a steady state, [11C]raclopride was injected using a bolus + constant infusion method[11]. Temporal variations in the [11C]raclopride signal were calculated using Equation 10. Approximately 10 min after CNO injection a significant increase in the temporal variations was observed in the striatum compared to vehicle-injected mice (paired Student's t-test, $n = 6$, $t = 2.858$, $df = 5$, $p = 0.035$) (Fig. 9a, b) with the left ventral striatum having the largest change of temporal variations.

In summary, our data support the theoretically predicted link between DA transient rates and variations of extracellular DA concentrations at high (0.5 Hz) and low (0.01 Hz) frequency. Activation of dopaminergic neurons significantly increases all three components: the rate of DA transients and the power of high and low frequency variations. This finding implies that all three components are measures of in vivo DA activity. While the detection of high frequency variations requires methods with high temporal resolution data acquisition, low frequency variations can be assessed with methods using lower temporal resolution. In particular low frequency variations induce detectable variations in the [11C]raclopride PET signal.

**DA release in response to food intake in humans.** By application of our method to human subjects who received either milkshake or a tasteless solution during the PET scan we found significant differences at the time of supply in reward-related regions of the DA system (ventral striatum ($p_{FWE,cluster} = 0.003$), habenula ($p_{FWE,cluster} = 0.0007$), substantia nigra/VTA ($p_{FWE,cluster} < 0.0001$), and pons (NTS; $p_{FWE,cluster} = 0.001$; highlighted in red in Fig. 10). Stimulation appears to induce detectable variations in the [11C]raclopride signal even in extrastriatal regions although there is less [11C]raclopride binding. Traditionally in PET analyses, data are integrated over time and each data point includes the history of kinetics from tracer injection up to this time point. In contrast, our method of analyzing signal variations from one-time frame to the next provides variations of [11C] raclopride as a function of time. Taking advantage of this aspect, we found—apart from differences during the time of milkshake/tasteless solution supply—a second-time interval with significant differences in DA activity at 15–20 min post-stimulation (highlighted in green in Fig. 10a, b). Again the pons was part of this

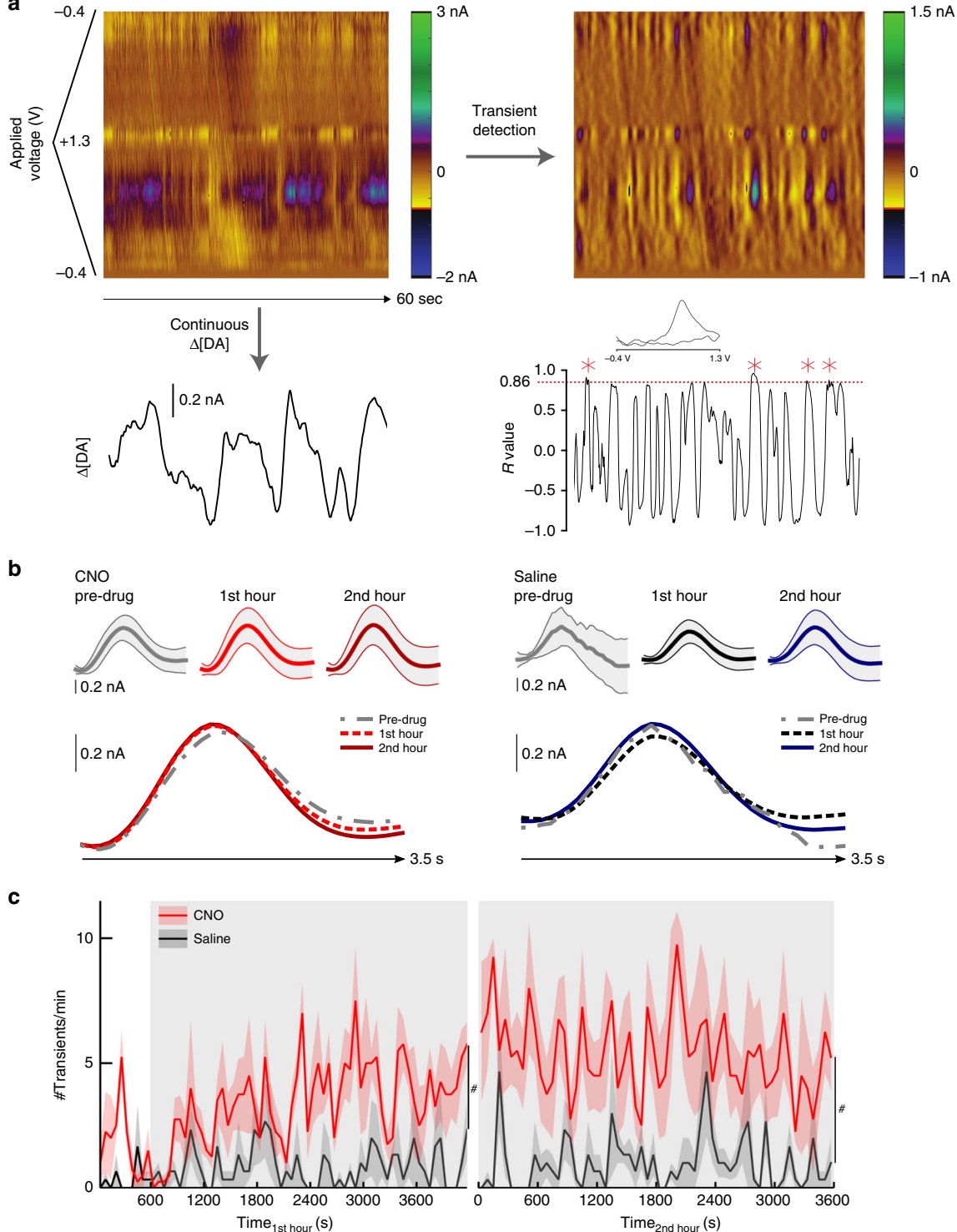

**Fig. 4** FSCV recordings of DA transients in mice. **a** Dopamine transients and levels in hM3D$_{Gq}$DAT mice. Top left panel shows the recorded current x applied voltage in a pseudocolor plot for a representative 60 s period in an hM3D$_{Gq}$DAT mouse after administration of CNO. Transients were detected by subtracting a moving 1 s window 1.5–0.5 s prior to each cyclic voltammogram (top right panel) and comparing the resulting cyclic voltammogram against an electrically-evoked dopamine template (bottom right panel). Spontaneous transients, identified as dopamine, are marked with a red asterisk; inset shows the average background subtracted cyclic voltammogram for these transients. Continuous dopamine levels were also extracted (bottom left panel) to allow comparisons between rates of transients and changes in dopamine at different timescales. **b** Average transient size (mean ± SD across all recorded transients) during the 15 min before drug administration, during the first hour, and second hour for the hM3D$_{Gq}$DAT CNO (left, $n = 4$) and hM3D$_{Gq}$DAT Saline (right, $n = 3$) groups. Transients were aligned on the peak r value and zeroed on the local minimum value in the preceding 1.5 s. **c** Spontaneous transients per minute ± SEM recorded in hM3D$_{Gq}$DAT mice before (white background) and after (gray background) CNO ($n = 4$) or Saline ($n = 3$) injection. Transients were counted in one-minute intervals. Data were analyzed via Two-way ANOVA, # = $p < 0.05$ for an overall main effect of treatment type

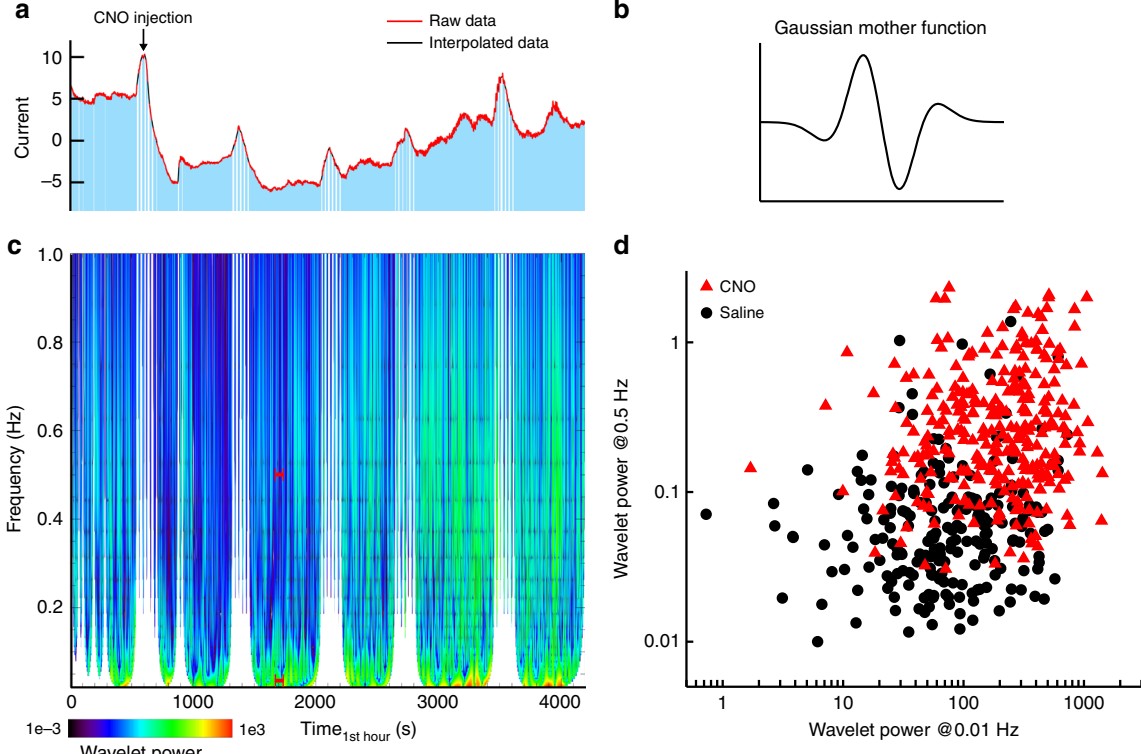

**Fig. 5** Wavelet power of continuous FSCV recordings. **a** Continuous FSCV data recorded in the ventral striatum of a single mouse. CNO was injected intraperitoneally at t = 600 s. **b** Gaussian mother function that was applied for the wavelet transform. **c** Contour plot of wavelet power calculated from the FSCV data shown in (**a**). Data gaps were excluded from the analysis together with the corresponding cone of influence (t±2*1/f). The red bars indicate a one-minute interval of low (0.01 Hz) and high frequency (0.5 Hz) power. **d** Power in the high and low-frequency band after CNO administration (red triangles = hM3D$_{Gq}$DAT+ CNO, n$_{CNO}$ = 4; black circles = hM3D$_{Gq}$DAT+ Saline, n$_{Saline}$ = 3). Each data point represents a one-minute interval

secondary delayed activation although at a different location (p$_{FWE,cluster}$<0.0001). Neighboring activation sites of the primary activation, also ventral posterior medial nucleus of the thalamus (VPM; p$_{FWE,cluster}$ = 0.018) and the dorsal striatum (p$_{FWE,cluster}$ = 0.0003) showed delayed secondary response to the milkshake. In rodents it was shown that vagus nerve-transmitted postingestive signals can induce DA response in the brain and that vagal nerve afferent terminals have D2 receptors[26,27]. This mechanism could be the origin of the delayed DA activity that we observe in our data. Note that in contrast to rDA the net [11C]raclopride uptake in the corresponding regions did not show any significant differences between milkshake and tasteless condition (Fig. 10c). These results demonstrate the power of our new method for the assessment of dopaminergic activity in humans. Further results of the human study can be found here[28].

## Discussion

To date, two main approaches have been introduced to detect DA release using PET. In the first approach, in baseline and stimulated conditions, the binding potential of the PET tracer is determined and differences are attributed to differences in released DA[7,12,29–33]. In the second approach, the kinetic model for the PET tracer was extended to account for dynamic changes of endogenous DA concentrations thereby allowing for transient DA release detection during PET data acquisition[10,11,14,17,34]. However, both approaches have in common that they require robust and long-lasting (> minute time scale) increases of DA concentrations, for example by pharmacological intervention prior to the PET measurement, to significantly reduce net

regional tracer binding, leading to detectable DA release events. In our method, we analyze temporal fluctuations instead of net reductions in regional tracer binding. The difference becomes evident when we compare time activity curves (TACs), resulting from net regional tracer binding, with rDA, calculated from the temporal PET signal fluctuations using the method described herein (Fig. 10a, c). Despite originating from the identical PET dataset, only rDA reveals two distinct time intervals with milkshake-induced activation, while the TACs do not indicate any difference between tasteless and milkshake conditions. Since the two approaches referenced above rely on net differences in the TACs, neither would be capable of detecting DA release. Supporting this observation, our FSCV data recorded in situ indicate that phasic dopamine release induces minute-by-minute fluctuations rather than minute-lasting elevations of DA concentrations implying that temporal PET signal variations are far more sensitive to detect DA release events than the net TAC. Thus, our method, in comparison to the currently utilized non-invasive detection methods, allows for the spatiotemporal assessment of physiologically relevant DA release events.

rDA, calculated as the ratio of local variations to the absolute local [11C]raclopride signal (Equation 10), is region-specific and cannot be compared between different regions, e.g. double rDA in the brain stem in comparison to the striatum does not mean that DA release rates in the brain stem are higher than in the striatum. But, as indicated by Fig. 1f, within a region rDA is proportional to the amplitude of DA release and can therefore be compared between subjects.

Our method for the detection of DA release is based on three premises: first, the [11C]raclopride signal measured by PET

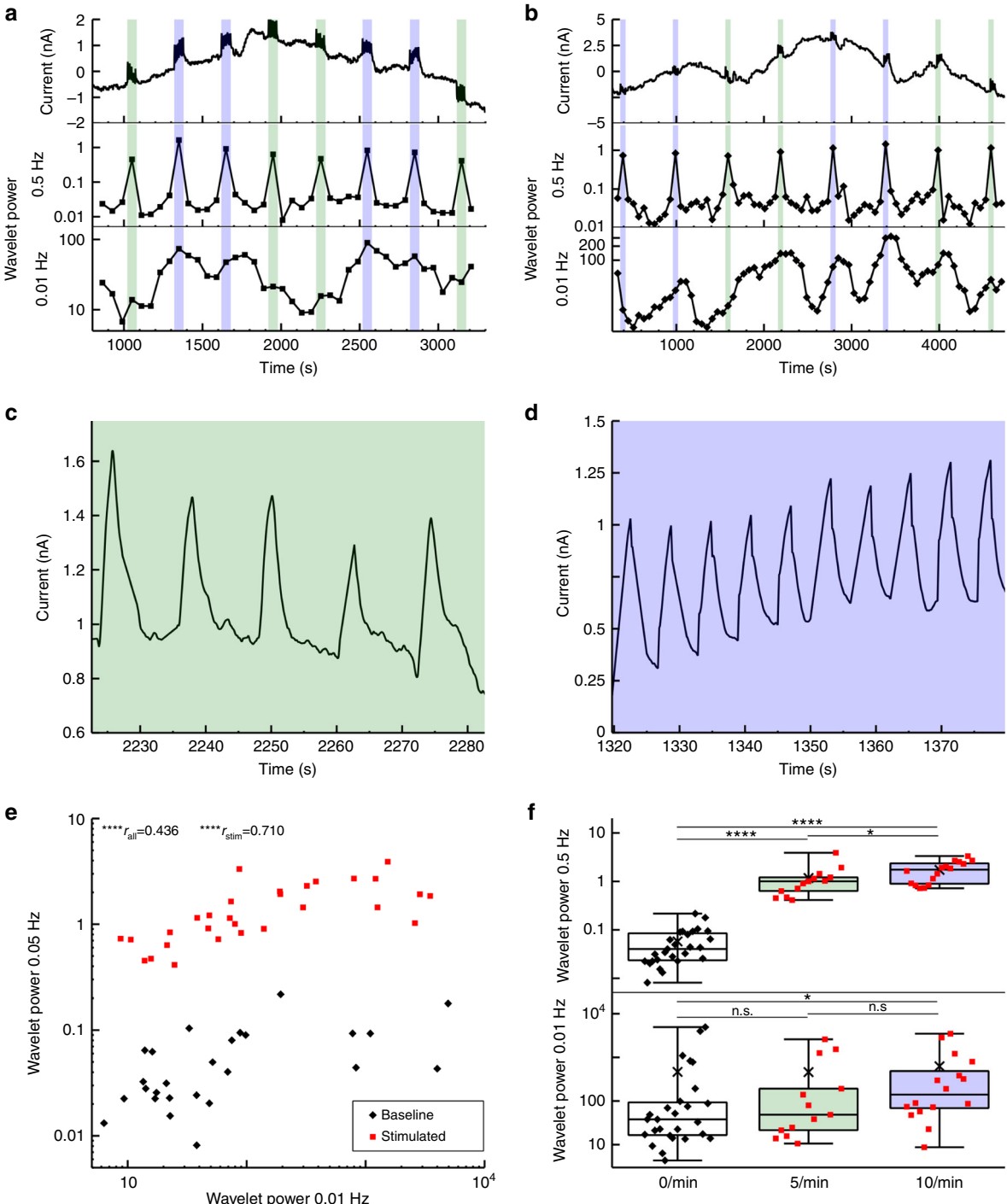

**Fig. 6** FSCV recordings of DA transients induced by electrical stimulation. (**a**, **b**, top trace) Continuous FSCV recordings from the ventral striatum of two mice. Five (green interval, **c**) or ten transients per minute (blue interval, **d**) were induced by electrical stimulation of the VTA. (**a** and **b**, mid and bottom trace) resulting wavelet power at 0.5 Hz and 0.01 Hz. **e** Correlation of wavelet power at 0.5 Hz and 0.01 Hz in stimulated one-minute intervals (red squares) and non-stimulated one-minute intervals taken from the center of non-stimulated time intervals (black diamonds, $n = 4$ mice). **f** Wavelet power in one-minute intervals as a function of the number of induced transients (median ± SEM, x = mean, whiskers indicate minimal and maximal values, $n = 4$ mice). Welch two-sample $t$-test: **** $= p < 0.001$, *** $= p < 0.005$, ** $= p < 0.01$, * $= p < 0.05$

originates predominantly from [11C]raclopride binding to extrasynaptic receptors, second, there is noticeable spillover of DA from synapses to the extrasynaptic extracellular space, and third, part of extrasynaptic extracellular DA is removed at a minute time scale. All three premises are well supported by the

literature. It has been shown that the majority of D2Rs are located outside and often distant from synapses[35,36]. Together with the fact that the synaptic volume is much smaller than the extrasynaptic extracellular volume, [11C]raclopride binds predominantly to extrasynaptic D2Rs. Spillover of DA from synapses

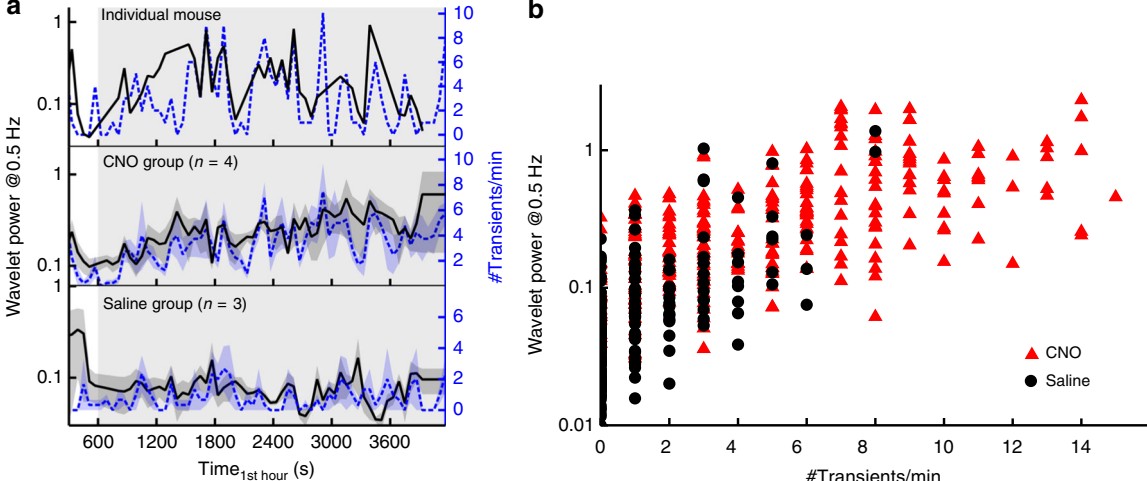

**Fig. 7** FSCV wavelet power at 0.5 Hz and transient rates in chemogenetically stimulated mice. **a** Comparison of the number of transients per minute (blue dashed lines) and the wavelet power at 0.5 Hz (black lines) calculated from continuously recorded FSCV data in one individual example mouse and in the CNO and Saline group (mean±SEM). The gray box indicates the time after CNO/ Saline injection. **b** Correlation of transient rate with the logarithm of the wavelet power at 0.5 Hz from all mice (red triangles = hM3D$_{Gq}^{DAT}$+ CNO, $n_{CNO}$ = 4; black circles = hM3D$_{Gq}^{DAT}$+ Saline, $n_{Saline}$ = 3; Pearson product moment correlations: $r_{CNO}$ = 0.64, $p_{CNO} < 10^{-16}$, $df_{CNO}$ = 296, $t_{CNO}$ = 14.26, $r_{Saline}$ = 0.63, $p_{Saline} < 10^{-16}$, $df_{Saline}$ = 215, $t_{Saline}$ = 11.75, $r_{combined}$ = 0.73, $p_{combined} < 10^{-16}$, $df_{combined}$ = 513, $t_{combined}$ = 24.44)

is the reason that DA transients can be reliably detected by FSCV although the electrodes cannot be positioned within the synaptic cleft. The concept of diffusive loss of DA from synapses has been elaborately discussed and is well established[18]. Moreover, minute time scale removal rates are often present in FSCV transient recordings – sometimes referred to as "hang-up". As discussed by A. Michael and colleagues "Evoked responses … with hang-up are absolutely commonplace."[20,37] Although in vitro experiments showed that part of the "hang-up" could be caused by adsorption of DA molecules on the electrode surface and could thereby be a methodological artifact, not all of the minute time scale DA clearance rates observed in vivo can be explained by this effect[38–40]. DA adsorption is related to exposure of the probe to high DA concentrations: the higher the peak and duration of DA concentration, the higher the DA adsorption. However, several observations show minute time scale DA clearance rates without the presence of a secondtime scale DA peak, which can accordingly not be caused by adsorption[41,42].

A potential mechanism that could explain the slow minute time scale component in the FSCV data could be the heterogeneous subcellular distribution of DATs. A fraction of the released DA diffuses into subcellular regions with low DAT expression and accordingly slow removal rates (Figure 2)[43,44]. In order to show that heterogeneous expression of DATs can cause minute time scale DA clearance rates, we performed model calculations that describe diffusion of DA through tissue with heterogeneous DAT expression. When we assumed homogenous DAT expression our model provided identical results to Cragg and Rice (Supplementary Figure 4A)[19]. Heterogeneous DAT expression, however, produced prolonged DA clearance rates (Supplementary Figure 4B). Furthermore, our calculations show that the contribution of the minute time scale component of a measured transient critically depends on the distance from the release site. Further away from the release site the slow component is more pronounced relative to the peak, while close to the release site the contribution of the slow component appears negligible relative to the peak in accordance with in vivo observations (Supplementary Figure 4C and D)[41]. Our model replicates another aspect of these in vivo data: when we compare the minute

time scale component recorded close to the release site with that recorded further distant, both are approximately identical (Supplementary Figure 4D). The accordance of our model results with in vivo observations indicates that transport and not diffusion determine the minute time scale dynamics and—given that DAT staining confirms the heterogeneous spatial distribution of DATs[44]—that heterogeneous DAT expression could be a potential mechanism to explain the minute time scale removal rates. Thus, the three prerequisites are in line with the current data on dopaminergic signal transduction.

Although the density of D2Rs in extrastriatal regions is much lower than in the striatum we were able to identify significant extrastriatal DA release. Several aspects promote the sensitivity of our method for detection of extrastriatal DA release. First, temporal variations in the signal relative to the total signal were analyzed instead of the absolute [11C]raclopride uptake, which makes the method more sensitive for the detection of variations in regions with low [11C]raclopride uptake. Second, although less DA is released in extrastriatal regions due the lower density of DA synapses, the major fraction of released DA is removed at a minute time scale due to the lower density of DATs. Therefore, the amplitudes of release-induced minute-by-minute variations in extracellular DA concentrations are of the same order of magnitude extrastriatally and in the striatum despite the difference in total amount of released DA (Figure 2)[42]. A detailed discussion of this topic can be found in the Supplementary Discussion.

We found with continuous FSCV recordings that CNO-treated hM3D$_{Gq}^{DAT}$ mice increased the rate of transients – although a low rate of spontaneous transients was detectable in non-activated mice. Typically, FSCV is applied to recording of transient phasic neurotransmitter release that is time-locked to direct stimulation of an upstream brain region or to a release-inducing behavioral task[41,45,46]. With continuous recording, spontaneous transients were identified by application of a DA template. Interestingly, we could show that wavelet power at 0.5 Hz strongly correlates with the transient rate. This means that transient rates can be derived from continuous FSCV data by application of a wavelet transform, which, in contrast to template application, does not require any assumptions or thresholds and

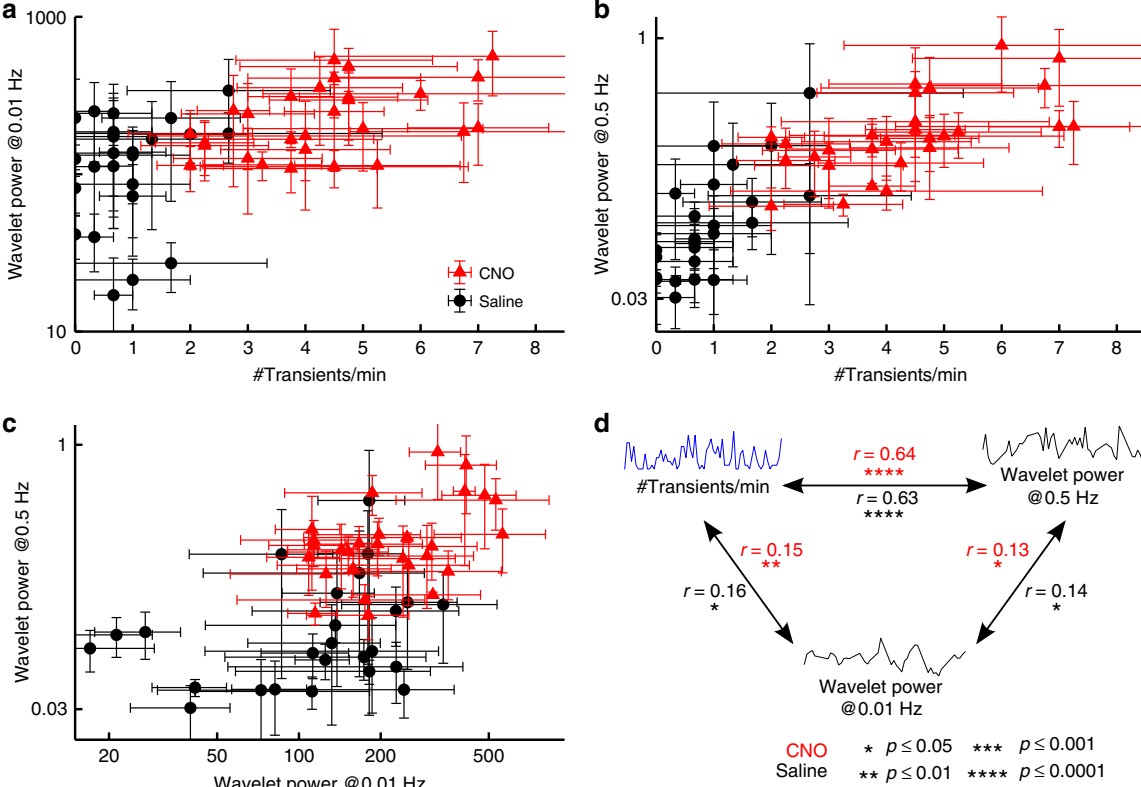

**Fig. 8** Correlations between transient rates and high as well as low-frequency power. **a–c** Correlations of the logarithm of low-frequency power (0.01 Hz) with transient rate (**a**), the logarithm of high-frequency power and transient rate (**b**), and the logarithm of low and high-frequency power (**c**). Data points are shown for each one-minute time interval as mean ± SEM for each group (red triangles = hM3D$_{Gq}$$^{DAT}$+ CNO, n$_{CNO}$ = 4; black circles = hM3D$_{Gq}$$^{DAT}$+ Saline, n$_{Saline}$ = 3). **d** Pearson product moment correlation coefficients, r, were calculated within each group (in red for the hM3D$_{Gq}$$^{DAT}$+ CNO group: df = 296, * = $p \leq 0.05$ ($t > 1.97$), ** = $p \leq 0.01$ ($t > 2.59$), *** = $p \leq 0.001$ ($t > 3.32$), **** = $p \leq 0.0001$ ($t > 3.94$); in black for the hM3D$_{Gq}$$^{DAT}$+ Saline group: df = 215, * = $p \leq 0.05$ ($t > 1.97$), ** = $p \leq 0.01$ ($t > 2.60$), *** = $p \leq 0.001$ ($t > 3.34$), **** = $p \leq 0.0001$ ($t > 3.96$)). All correlations were statistically significant

can therefore be applied more easily. Thus, wavelet transform is a powerful tool to extract changes of DA levels from continuous FSCV data.

Moreover, we found correlations between high and low-frequency wavelet power within the continuous data indicating that minute-by-minute DA levels might be causally linked to second-by-second levels. This finding agrees with another experimental FSCV study where DA levels were continuously recorded in the nucleus accumbens. The authors conclude that extracellular DA largely arises from phasic DA release[47]. The causal link between minute-by-minute and second-by-second DA levels potentially challenges functional segregation of the two processes: If minute-by-minute levels are just a consequence of second-by-second release, evoked functions are presumably related to each other. In other words: if phasic DA release determines the tonic DA level, the functional consequences of both might also be related[48]. (For further discussion see Berke[49]).

The relation of synaptic release and low-frequency variations of extracellular concentrations is based on fundamental principles (diffusion, transport)[18] and should theoretically not be limited to dopamine but should also apply to other neurotransmitters. However, whether conditions are suitable (signal to noise ratio, life time, etc.) to produce detectable signals of neurotransmitter activity, will have to be evaluated individually for each neurotransmitter system. Nevertheless, the method provided here not only opens novel avenues to study temporal and local dynamics of dopaminergic transmission in human, but potentially for a whole range of other neurotransmitters

## Methods

**Ethical approval**. All animal procedures were conducted in compliance with protocols approved by local governmental authorities (Bezirksregierung Köln) and were also in accordance with NIH guidelines for animal research. FSCV experiments were conducted under the auspices of the UK Home Office laws for the treatment of animals under scientific procedures and of the University of Oxford ethical review board.

All subjects included in the human PET study gave written informed consent prior to study, which was approved by the local ethics committee of the Medical Faculty of the University of Cologne (Cologne, Germany).

**Genetic mouse models**. The Rosa26CAGSloxSTOPloxhM3D$_{Gq}$ and dopamine transporter-Cre recombinase (DAT-Cre) expressing mice are described elsewhere[50,51]. All animal lines were maintained on C57BL/6N backgrounds. To generate the experimental model, homozygous Rosa26CAGSloxSTOPloxhM3D$_{Gq}$ (hM3D$_{Gq}$) females were crossed to DAT-Cre males, to generate mice with heterozygous expression of the hM3D$_{Gq}$ specifically in dopamine neurons. Non-transgenic littermate controls were used in the behavioral experiments (referred to as hM3D$_{Gq}$$^{WT}$). In all other experiments Cre positive animals were used (referred to as hM3D$_{Gq}$$^{DAT}$). Mice were housed at 22 °C-24 °C with a 12 h light/12 h dark cycle. Animals had *ad libitum* access to food and water in the home cage at all times. All experiments were performed in adult male mice (age: 13–37 weeks, body weight: 24.6–47.9 g).

**CNO administration**. Clozapine-*n*-oxide (CNO) was purchased from Sigma (Cat. No. C0832–5MG). A 5 mg/mL stock solution was made using DMSO (Sigma). 32 μL aliquots were stored at −20 °C, and on individual test days, a working solution of 0.03 mg/mL was generated using sterile saline (0.9%; Aquapharm). All injections were made intraperitoneally. Sterile Saline was used for vehicle injection in all experiments unless otherwise stated. The vehicle control consisted of 0.6% DMSO in saline.

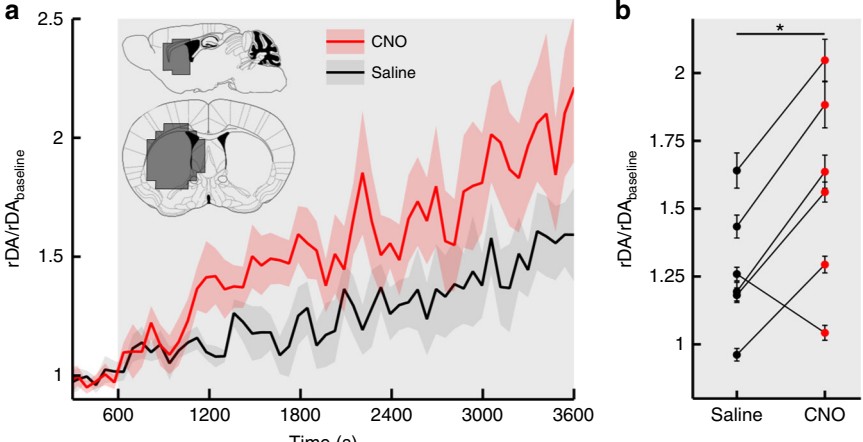

**Fig. 9** Dopamine release assessed by [11C]raclopride-PET in hM3D$_{Gq}$$^{DAT}$ mice. **a** Dopaminergic activity changes temporal variations of the [11C]raclopride signal (*rDA*, mean±SEM). In *n* = 6 mice *rDA* was increased in the left striatum after chemogenetic activation with CNO (red line) in comparison to Saline injection (black line) at *t* = 600 s. The gray background indicates the time after injection. **b** Average *rDA* (mean ± SEM) after *t* = 600 s in each mouse after saline and CNO injection. (Paired Student's *t*-test: *n* = 6, *t* = 2.858, df = 5, * = *p* = 0.035)

**Open field test**. hM3D$_{Gq}$$^{DAT}$ and hM3D$_{Gq}$$^{WT}$ mice were acclimated for 5 days to daily injections prior to the test day. On the test day, animals received an injection of CNO (0.3 mg/kg, Sigma) or sterile saline (10 µl/g BW) and were placed immediately into open field chambers (27.3 cm × 27.3 cm × 20.3 cm; Med Associates, VT). The animals were monitored for 120 min in the chambers using infrared beam breaks in the X, Y and Z planes and data was collected in 10-minute bins. Post analysis of locomotor and rotational behaviors were assessed using MedActivity Software (Med Associates, VT).

**Electrode implantation surgery**. Mice were anesthetized using isoflurane (3–4% for induction and 1.0–1.5 % for maintenance) and given 5 mg/kg meloxicam (Metacam) and 0.7 ml glucosesaline (0.5 % in 0.9 % saline; Aquapharm) sub-cutaneously (note, no opioid analgesic was given to avoid any potential interactions with dopamine systems). After induction, the head was shaved and secured in a stereotaxic frame. Body temperature was maintained at 36–37 °C with the use of a homeothermic heating blanket. Corneal dehydration was prevented with application of ophthalmic ointment (Lacri-Lube, Allergan, UK). The head was then cleaned with dilute Hibiscrub and Reprochem (diluted 1:20 in water), and a local anesthetic, bupivacaine (2 mg/kg; AstraZeneca), was administered under the scalp. The skull was then exposed and holes were drilled for an Ag/AgCl reference electrode, an anchoring screw, a recording electrode and a stimulating electrode. After the reference electrode was secured in place using dental cement (Kemdent, Swindon, UK), a custom-made carbon fiber microelectrode was attached to a voltammetric amplifier and lowered toward the dorsal nucleus accumbens (NAc) core (AP: +1.4, ML: 0.75, DV: −3.5 to −4.25 from skull) followed by a 2-channel untwisted stimulating electrode (PlasticsOne) to the ipsilateral ventral tegmental area (VTA) (AP: −3.5, ML: 0.35, DV: −4.0 to −4.55 from brain), and the recording process commenced (see below). The reference electrode and anchoring screw were positioned contralateral to the carbon fiber and stimulating electrodes. The mouse was given additional boluses of 0.7 ml glucosesaline ~every 3 h for the rest of the surgery.

**Fast-scan cyclic voltammetry recordings**. Recordings of in vivo nucleus accumbens (NAc) core dopamine levels were made under anesthesia using FSCV. The potential applied to the carbon fiber was ramped from −0.4 V (vs Ag/AgCl) to +1.3 V and back at a rate of 400 V/s during a voltammetric scan and held at −0.4 V between scans. This happened at a frequency of 60 Hz for an initial 20-minute period in order to condition the electrode, after which scan rate was reduced to 10 Hz for the rest of the experiment and dopamine detection commenced. Electrical stimulation was applied using an isolated current stimulator (DS3, Digitimer). Stimuli were generated and recordings collected using Tarheel CV (National Instruments). The positions of the recording and stimulating electrodes were optimized by moving them to find the maximal changes in dopamine that could be detected after stimulation (50 × 2 ms monophasic pulses, 200 µA current, at 50 Hz).

Once this was achieved, the main experiment was started to determine the effect of chemogenetic activation of dopamine neurons on patterns of NAc core dopamine release. This consisted of two situations: (1) monitoring spontaneous changes in dopamine levels in the absence of external stimulation, and (2) examining evoked dopamine release after electrical stimulation of the VTA. For the latter, we used 5 different stimulation parameters (2 recordings with each, 3 min

between stimulations): (i) 20 pulses 100 µA, (ii) 30 pulses 100 µA, (iii) 30 pulses 150 µA, (iv) 40 pulses 150 µA, and (v) 50 pulses 200 µA. This was performed 3 times: before mice were given an intraperitoneal injection of either CNO (0.3 mg/ kg body weight) or vehicle (0.6% DMSO in sterile saline), 1 h after injection, and at the end of the experiment (~145 min after injection). For the middle set of stimulations, only parameters (ii), (iv) and (v) were used. With the exception of the first mouse that had a slightly different set of stimulation parameters the protocol described was used for all mice.

To examine spontaneous changes in dopamine, we continuously monitored dopamine levels under anesthesia, first for 10 min prior to injection of either CNO or vehicle, then for 60 min after injection, and then finally for a further 60 min after the middle set of electrical stimulations.

Additional continuous FSCV recordings were performed in the ventral striatum of four wild-type mice. In one-minute time intervals DA release was induced by electrical stimulation of the VTA at a rate of either 5 or 10 transients per minute followed by a resting time interval of either 4 min (mouse 1) or 9 min (mouse 2, 3, and 4). The parameters of the electrical stimulation were adjusted to 50 Hz, 4–7 pulses, and 100–200 µA in order to obtain transients with an average amplitude of ~1–2 nA. Recordings of mouse 3 and 4 displayed periods where the chemometric model failed. Affected time intervals were excluded from the analysis. Note, that all reported significances remain significant if these intervals are included in the analysis.

**Voltammetry data analysis**. Voltammetric analysis was carried out using custom-written scripts in Matlab. All data were low-pass filtered at 2 kHz. In order to characterize the rate of spontaneous dopamine transients in each mouse, we first subtracted the average current recorded between 1.5–0.5 s before the target cyclic voltammogram to account for large changes in capacitance current. We then looked for periods when the cyclic voltammograms recorded over the course of the experiment correlated with a correlation coefficient of R ≥ 0.86 with a dopamine "template" derived by electrically stimulating the VTA before the experiment began (Daberkow, 2013; Cheer, 2004). The numbers of transients per minute before and after CNO or vehicle injections were then compared.

To extract an estimate of changes in dopamine levels over time across the session, a principal component analysis was performed using a standard training set of stimulated dopamine release detected by chronically implanted electrodes, with dopamine treated as the first principal component among other unrelated electrochemical fluctuations such as changes in pH. For this analysis, we divided the data into non-overlapping 30 s bins and, for each, subtracted the average current recorded over the initial 1 s in each bin. Given that it is only possible to derive a relative and not an absolute measurement of dopamine levels using FSCV, the extracted dopamine in each bin were combined by assuming that the first recorded value in bin N+1 continued relative to the last time point in bin N.

**Wavelet transform of continuous FSCV data**. A wavelet transform decomposes the signal into harmonic functions of different frequencies but in contrast to the Fourier transform these harmonic functions have a finite duration. Thereby, a wavelet transform does not lose the temporal information of the signal[25]. The wavelet power calculated as the square of the wavelet coefficients gives the power in variations of extracellular DA levels as a function of frequency and time.

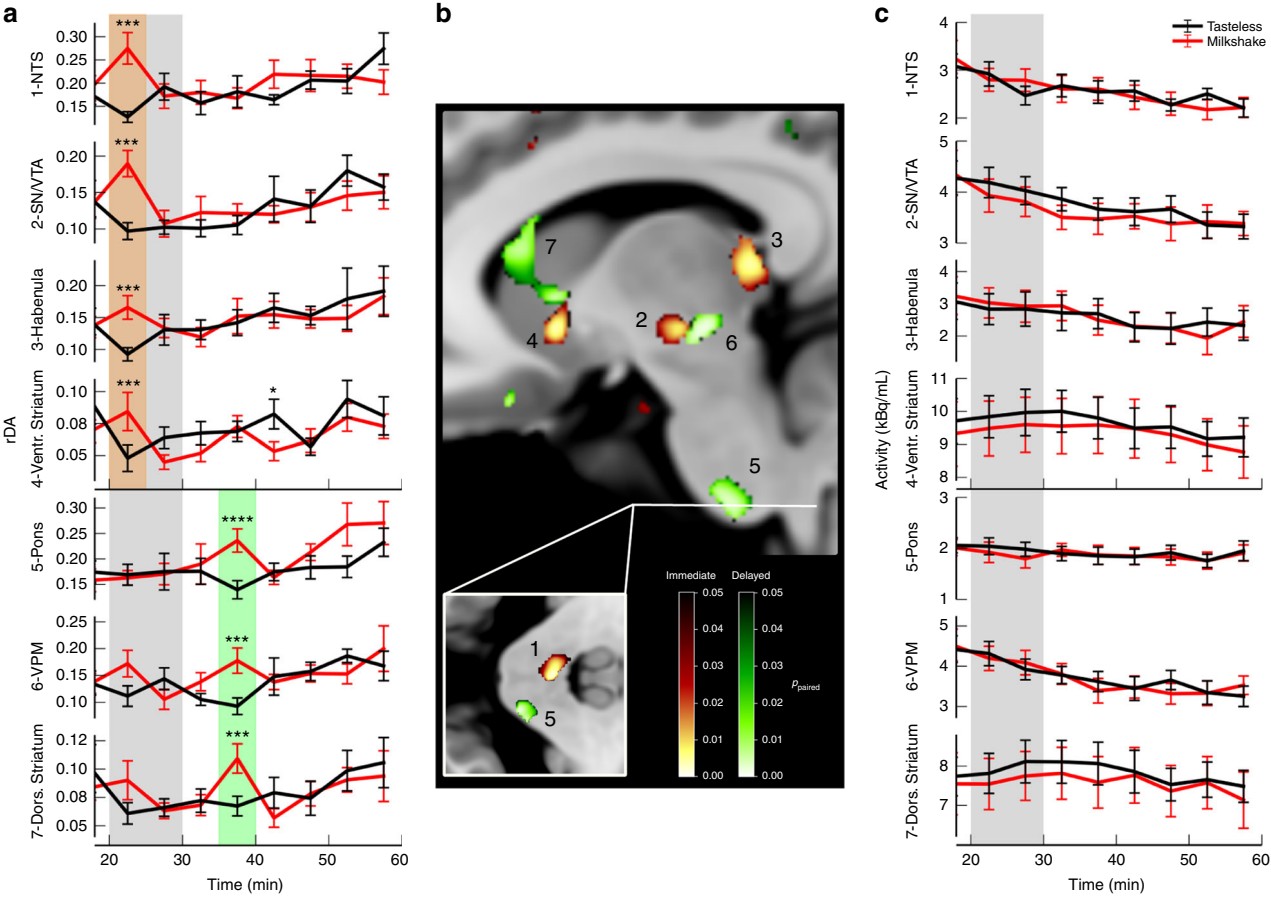

**Fig. 10** Food-induced DA release in humans. During two separate PET imaging sessions human volunteers ($n = 10$) received either a tasty milkshake or a tasteless solution starting 20 min after [11C]raclopride injection for 10 min. **a** Time course of *rDA* in regions with significant differences between milkshake (red) and tasteless (black) condition in the 20–25 min (orange box) or 35–40 min (green box) time interval. The gray box indicates the time of milkshake/tasteless solution administration. **b** Locations of regions with significant differences of *rDA* between milkshake and tasteless condition in the 20–25 min (red color scale) or 35–40 min (green color scale) time interval. **c** Time activity curves in the regions indicated in (**b**) showing no detectable differences between the tasteless and milkshake condition. Identified regions are 1: nucleus of the solitary tract (NTS), 2: substantia nigra/VTA, 3: habenula, 4: ventral striatum, 5: pons, 6: ventral posterior medial nucleus of the thalamus (VPM), and 7: dorsal striatum. All data are represented as mean±SEM, paired Student's *t*-test: df = 9, **** = $p < 0.001$ ($t > 4.78$), *** = $p < 0.005$ ($t > 3.69$), * = $p < 0.05$ ($t > 2.26$) (uncorrected)

For the analysis of high and low-frequency variations in the FSCV data a wavelet transform with a Gaussian mother function (3. order) was applied. Since continuous data is needed for the wavelet transform missing points in the data were interpolated. Using a cone of influence the wavelet data of these segments were removed from the resulting spectra ($t\pm2*1/f$). Additionally, to avoid edge effects the borders of the wavelet transform were also removed ($t_{min}+4*1/f$ and $t_{max}-4*1/f$). The power in the wavelet spectrum at 0.01 and 0.5 Hz was averaged in windows of 1 min to yield time courses of wavelet power at 0.01 Hz and at 0.5 Hz for each animal. Correlation analyses were restricted to time intervals that contained continuous data in all three parameters (#transients/minute, wavelet power at 0.01 Hz and at 0.5 Hz).

**Synthesis of [11C]raclopride**. [11C]-O-methylraclopride was prepared according to Langer et al.[52] with the following modifications: (1) 2–2.5 mg precursor desmethylraclopride•HBr was used instead of free base, (2) 350 µl DMSO together with 6 µl 5N NaOH was used instead of acetone as reaction solvent, (3) the reaction mixture was heated at 80 °C for 4 min, (4) the eluent was removed after semi-preparative HPLC separation by solid phase extraction with a Waters OASIS MCX 1cc cartridge, eluted with 500–700 µl ethanol 96%/NH3 (95:5) and 2 ml 0.9% NaCl solution, and (5) formulation by adding another 6.5 ml 0.9% NaCl solution and 1 ml 125 mM sodium phosphate buffer, resulting in a solution with pH 5–7. The specific activity at time of injection was 22.8 ± 9.0 GBq/µmol.

**Mouse PET data acquisition**. Dynamic PET data were acquired using a combined preclinical PET/CT scanner (Inveon, Siemens). For each scan session of 60 min two animals were placed on a water-heated mouse carrier with stereo-tactic holders (Medres). During the procedure mice were anesthetized with ~2 % isoflurane vaporized in 1.0 L/min of oxygen-nitrogen gas (30 % $O_2$ / 70 % $N_2$). At the start of the PET data acquisition the animals received a bolus-plus-constant-infusion injection of 10.5 ± 2.6 MBq of [11C]raclopride via the tail vein: a bolus of 80 µl was injected in one minute, followed by additional 120 µl injected via constant infusion until the end of data acquisition using programmable syringe pumps („Genie" Kent, Kent Scientific Corp., Torrington, CT). 10 min after the start of the PET scan the mice either received CNO (0.3 mg/kg body weight) or sterile saline (10 µl/kg body weight) intraperitoneally. Each animal was measured twice in a randomized order once receiving CNO and once saline. Following the PET scan the animals were automatically moved into the CT gantry and a CT scan was performed (180 projections/360°, 200 ms, 80 kV, 500 µA). CT data were used for attenuation correction of the PET data and the CT image of the skull was used for image co-registration.

**Mouse PET data processing**. PET data were histogrammed in 60 time frames of 1 min, Fourier rebinned and after correction for attenuation and decay, images were reconstructed using the MAP-SP algorithm provided by the manufacturer. The images were co-registered to a reference mouse brain CT by rigid body transformation using the imaging software VINCI[53]. Parametric images were calculated using Equation 10 (see detailed description below, procedures written in IDL and C).

Using a 3D mouse atlas constructed from a 2D mouse brain atlas[54], an anatomical volume of interest (VOI) of the left striatum was drawn. This region was analyzed since FSCV data were acquired from this region.

**Human PET**. Although invasive methods such as FSCV have been applied to measure task-related neurotransmitter release in the human brain[55–57], PET measurements are non-invasive and therefore much more readily translatable to humans. In contrast to mice, human subjects do not require anesthesia during the PET scan and natural non-pharmacological types of stimulation, such as food, can be applied as a stimulus to induce DA release. In order to analyze food-related reward signaling, we applied the method to healthy human volunteers who received a milkshake during [11C]raclopride-PET data acquisition. Human subjects ($n = 10$) were monitored in two different conditions: in one session they received a palatable milkshake and in the other session a tasteless solution. Following the method introduced here, maps of DA activity were calculated in time intervals of five minutes. Voxelwise statistical testing was performed (paired Student's t-test) between the two conditions to identify locations with stimulation-induced changes in dopaminergic activity. Here we discuss only exemplary aspects of that data as a proof of principle and to demonstrate the power of the new approach. The full-length results of this study can be found here[28].

Subjects were ten healthy, male, normal-weight (BMI: 25.73 ± 2.67, age: 57.1 ± 10.55) and non-smoking volunteers recruited from a preexisting database of Max Planck Institute for Metabolism Research. No history of neurological, psychiatric or eating disorders were present. Further exclusion criteria were special diets, lactose intolerance, diabetes, the participation in a previous PET study and a score higher than 12 in the Beck Depression Inventory (BDI II)[58].

**Human PET data acquisition**. Two PET scans were performed in a randomized order with the subjects receiving either milkshakes or a tasteless and non-nutritive solution (potassium chloride/sodium bicarbonate) during the scan. A HRRT Siemens PET gantry with a spatial resolution of ~2.5 mm FWHM was used. The head of the subjects was fixed by an inflatable helmet to prevent motion during data acquisition. Data for attenuation correction was acquired by performing a ten-minute transmission scan using a rotating germanium-68/gallium-68 source. Afterwards, [11C]raclopride injection started and emission data were acquired for the following 60 min 70 % of the [11C]raclopride (220–370 MBq) was applied as bolus within a minute and 30 % was constantly infused during the remaining 59 min using a programmable syringe pump (Perfusor compact, Braun, Melsungen). The food stimulus started 20 min after the start of the data acquisition, when steady-state conditions were reached, and lasted for 10 min During this time either milkshake or tasteless solution was delivered to the tongue tip of the subjects via a teflon mouthpiece that was attached to the gantry. For further details see ref. [28].

**Human PET data processing**. The acquired emission data were corrected for attenuation and scatter. PET images were reconstructed in 12-time frames of 5 min duration using three-dimensional ordinary Poisson ordered subset expectation maximization (OP-3D-OSEM) including the modeling of the system's point spread function (PSF). The resulting images were smoothed using a 10 mm Gaussian filter using the imaging software VINCI[53]. The smoothed PET images were co-registered to an additionally acquired individual anatomical T1-weighted scan by rigid body transformation. The MR scans were normalized into the Montreal Neurological Institute (MNI) stereotactic space using a non-linear transformation algorithm (VINCI). The obtained transformation matrix from this step was subsequently applied to the coregistered PET images to transform them into the MNI-152 standard space.

**Parameter for measuring temporal dynamics of [11C]raclopride**. From the [11C]raclopride PET images parametric maps were calculated. In the manuscript we have shown that variations of the [11C]raclopride signal are related to variations of D2R-bound DA in the ECS. Part of the temporal variations in the [11C]raclopride signal is due to noise. However, since the noise level does not abruptly change during the measurement, short-term changes in the temporal variations are presumably caused by changes in DA activity. Temporal variation of the [11C]raclopride signal at time $t_n$ and location (image voxel) $i,j,k$ can be calculated as:

$$rDA \equiv \frac{\Delta R_{ijk}}{R_{0,ijk}}(t_n) = \frac{1}{R_{0,ijk}} \sqrt{\frac{1}{N_{sum}} \sum_{u=i-d_i}^{i+d_i} \sum_{v=j-d_j}^{j+d_j} \sum_{w=k-d_k}^{k+d_k} (R_{uvw}(t_n) - R_{uvw}(t_{n-1}))^2}$$

(10)

$$R_{0,ijk} = \frac{1}{N-ns} \frac{1}{N_{sum}} \sum_{n=ns}^{N} \sum_{u=i-d_i}^{i+d_i} \sum_{v=j-d_j}^{j+d_j} \sum_{w=k-d_k}^{k+d_k} R_{uvw}(t_n)$$

(11)

In order to reduce noise in the parameter for temporal variations of the tissue [11C]raclopride signal we here calculate the average of the absolute difference between the signal at time $t_n$ and $t_{n-1}$ in a region that includes the box $i-d_i,i+d_i,j -d_j,j+d_j,k-d_k,k+d_k$. The square root term is a measure for $\Delta R(t_n)$ the average change from time $t_{n-1}$ to time $t_n$ of the [11C]raclopride signal in the box centered at $i,j,k$ ($N_{sum}$ is the number of voxels in the box). $R_0$ is calculated in Equation 11 as the average signal in the box after the quasi steady state has been reached at $t_{ns} =$

20 min after bolus injection. The size of $d$ reduces the spatial resolution but also reduces spatial noise in the parameter. The choice of $d$ depends on the system used for data acquisition and on the quality of the data. For mouse data acquired we used $d_i = d_j = 2$ and $d_k = 1$ (asymmetric voxel size), for the human data we used $d_i = d_j = d_k = 2$.

It is important that $rDA = \Delta R(t)/R_0$ is calculated from the square root of the sum-of-squares and not just from the sum of differences (Equation 10). From theory, the absolute value of temporal changes but not the direction of change (increase or decrease) is a measure for DA activity.

$\Delta R(t_n)$ is calculated from the difference in [11C]raclopride signal at time $t_n$ and time $t_{n-1}$ and therefore depends on the duration of time frames ($\Delta t = t_n - t_{n-1}$) of the PET data. Although shorter time frames provide higher temporal resolution, Fig. 1f indicates that shorter time frames (i.e. higher frequencies) are less sensitive to variations of DA and apart from that include more noise inherent in the measurement procedure. For the mouse data we used $\Delta t = 1$ min, for humans we used $\Delta t = 5$ min

$rDA$ is a measure for local temporal variations of the [11C]raclopride signal ($\Delta R$) relative to total local [11C]raclopride signal ($R_0$). $rDA$ is thereby a measure for local DA activity and is thus comparable between subjects but it cannot be compared between regions. Since D2R density in extrastriatal regions is lower than in the striatum there is consequently less total [11C]raclopride binding. Higher $rDA$ in an extrastriatal region than in the striatum does not necessarily mean that more DA was released in the extrastriatal region. But following the time course of $rDA$ within a region indicates temporal variations of DA activity.

**Statistical testing**. For behavioral studies, Three-Way ANOVA analysis of data collected over time was conducted using JASP Version 0.8.3.1 (University of Amsterdam), followed by Two-way ANOVA post-hoc analysis at individual time points corrected for false discovery rate using GraphPad Prism version 7.0c for Mac OS X, (GraphPad Software, California, USA, www.graphpad.com). Two-way ANOVA followed by Bonferroni post-hoc analysis of locomotor behavior totals and rotational behavior was performed using GraphPad Prism version 7.0c. P-values<0.0.5 were considered significant. For FSCV analyses, Pearson product moment correlations were calculated using R. For testing differences between groups unpaired Student's t-test was performed. P-values <0.05 were considered significant. In mouse PET studies differences between activation and baseline were determined by performing a paired Student's t-test. P-values <0.05 were considered significant. In the human PET study voxel-wise independent paired Student's t-tests of parametric images were performed between milkshake and tasteless scans for each time frame. Clusters with statistically significant differences (p-value <0.05) were corrected for multiple comparisons by calculating family-wise error rates on cluster level[59]. All statistical tests in this work were two-sided.

**Code availability**. Custom computer code used to generate the results of this study is available from the corresponding author upon reasonable request.

**Reporting summary**. Further information on experimental design is available in the Nature Research Reporting Summary linked to this article.

## Data availability
The data that support the findings of this study are available from the corresponding author upon reasonable request.

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

## Acknowledgements

M.EW. and C.K. were supported through a Wellcome Trust Senior Research Fellowship to M.E.W. (202831/Z/16/Z). H.B., M.T. and J.C.B were supported by of the German Research Foundation in the Transregional Colloborative Research Center 134. M.T. and J.C.B were supported by the German Centre for Diabetes Research. [11C]raclopride was provided by the radiochemistry lab of Prof. Dr. Bernd Neumaier. We kindly thank N.G. Larsson (Max Planck Institute for Biology of Ageing, Cologne, Germany) for providing Dat-Cre mice.

## Author contributions

Conceptualization, R.L. and H.B.; Methodology, R.L., A.L.C and H.B.; Software, A.L.C., T.J.P. and H.B.; Validation, H.B., R.L., A.L.C., and M.W.; Formal analysis, R.L., A.L.C., C. K., T.J.P., L.M.B., and H.B.; Investigation, H.B., R.L., A.L.C., M.W., C.K., L.M.B., and S.E.

T.; Writing—Original Draft, H.B.; Writing—Review & Editing, R.L., A.L.C., and H.B.; Visualization, H.B., A.L.C., and R.L.; Supervision, M.T., J.C.B., M.W., and H.B.

## Additional information

**Competing interests:** The authors declare no competing interests.

