## [Peer Review File · Nature Communications]

Reviewers' comments:

Reviewer #1 (Remarks to the Author):

The manuscript by Lippert et al presents an innovative method of analyzing PET data that purports to harness the dynamic changes in the PET signal to provide information about underlying fluctuations in dopamine release. To support their claims the authors combine modelling with fast-scan cyclic voltammetry in mice and PET experiments in mice and humans. I have direct experience with voltammetry but not with PET and I have limited modeling experience. Thus, I feel better equipped to judge some of their claims than others. However, despite my limited experience I found the report easy to read and gain a decent understanding of the theory, the experiments, and the potential significance. The basic idea, as far as I can ascertain, is that second-to-second fluctuations in dopamine release are well-correlated with minute-to-minute fluctuations and this correlation allows the dynamic PET signal, analyzed in this novel way, to act as a proxy for the changes in release that are usually measured with invasive techniques such as voltammetry. This paper and the technique it introduces could be potentially of great interest to many and may shift the way PET data is dealt with, analyzed, and interpreted in the future. I do have a number of questions and suggestions that should be taken into account by the authors and editor.

Major comment(s)

My main point is that while I am fairly well convinced that this new method of analysis may pick up differences in regional dopamine function, as the voltammetry data is taken from a single type of manipulation (CNO-evoked transients), I am wary of the interpretation becoming too broad. For example, I do not believe enough has been done to firmly state that the difference reported in the PET signal is due to an increase in transient number, or even dopamine release events.

For example, the manipulation that the authors use may produce a certain set of results (e.g. the correlations between transient number, and high and low frequency wavelet power) that might not exist if a different method of altering dopamine function had been used. This is pertinent because of the possibility that the slow ramping of the signals in this model could potentially lead to artifacts in the wavelet transformation at low powers.

The study would have been significantly strengthened if the authors had managed to increase (and even suppress) the number of dopamine transients in a more intermittent, on-off fashion, for example, by using optogenetics or a pharmacological manipulation with a short half-life.

Another related issue involves the time course of transients. Within the striatum, duration of transients varies significantly due to differences in DAT expression. Other extrastriatal brain regions, have vastly different levels of DAT, which will affect transient duration. In addition, different release patterns of dopamine may also exist depending on the firing of projecting cells and terminal mechanisms. It may be significant that I am not aware of spontaneous dopamine transients being recorded with voltammetry in any extrastriatal region. What implications does this have for the authors' PET findings in extrastriatal regions.

Minor

1. To what extent will PET experiments that have already been conducted be able to benefit from this new analysis method? The authors mention that between-region analysis is uninformative and so experimental design and data collection will need to have been conducted in a certain way. This is not a criticism of the study but something that the authors may wish to comment on in a little more detail.

2. Equally, the authors' claims about the correlation between low and high frequency wavelet power informing the long-running discussion about phasic vs. tonic dopamine is intriguing and attractive. They could think about expanding this a little.

3. More information should be provided on the mouse line used – e.g. body weights, have they been previously published, any other phenotypic observations? In addition, it should be noted that all DAT-expressing neurons will be affected in this model, not just midbrain dopamine neurons.

4. Showing the stimulated release data would be useful addition – is it the same between genotypes?
5. Could the authors make it easier to see which trace is which in panel 3B - the overlays with the standard error are difficult to interpret. One possibility would be to show each trace side by side with its standard error and then an overlay with just the averages?
6. The authors state that “[Despite] minor differences in spontaneous baseline activity apparent in the first 5 minutes of baseline recording...”. This difference actually seems quite significant. What’s the interpretation? Also, were these data included in their ANOVA?
7. In the individual mouse trace (Fig 5A, top) comparing transient rate and power in the 0.5 Hz band, although the signals tend to be correlated, there are times at which these signals deviate. Do the authors have any hypothesis as to what causes these deviations? Could they be related, for example, to a change in transient properties (e.g. transients with double-peaks or a longer duration)?
8. In Fig 7 it would be good to show where in the brain these data are taken from, for example, with a simple schematic.
9. In Fig 8, the caption should more clearly state what is going on in each panel. It describes everything quite well but is difficult to quickly find what is in panel C, for example.

Reviewer #2 (Remarks to the Author):

The manuscript by Backes and colleagues outlines a novel method of analyzing PET signals to greatly increase the temporal resolution of the technique. They propose a mathematical model to relate slow changes in D2 receptor-bound raclopride levels to fast increases in extracellular dopamine due to synaptic activity. Fast scan cyclic voltammetry (FSCV) in anesthetized mice was used to validate the model. The continuous FSCV methods used to monitor rapid spontaneous synchronous dopamine release events (transients) appear to be appropriate and rigorous, and the analysis of the FSCV data is both elegant and novel. Chemogenetic activation of dopamine neurons with CNO is used to increase extracellular dopamine, and both FSCV and the new PET method are able to measure the changes. CNO has a fairly rapid onset, on the order of several minutes, but has a slow offset (30 min-2 hours depending on dose), and therefore may not be the best choice for testing the limits of the PET model. Repeated electrical stimulations of VTA neurons for 1 second or less might provide a better challenge to test the authors’ claim that minute-by-minute fluctuations of the PET signal relate directly to subsecond dopamine release events. Also, the manuscript needs to include an in-depth discussion of the work of Kishida and Montague describing FSCV measurements of real-time spontaneous dopamine and serotonin signals in human brain during decision-making tasks (Kishida et al., 2016; Moran et al., 2018), and the relationship between rapid dopamine signaling and BOLD responses in human subjects (Lohrenz et al., 2016).

Reviewer #3 (Remarks to the Author):

In this paper, Lippert and colleagues evaluate a novel method for measuring changes in synaptic dopamine release using PET and the D2/D3 antagonist radioligand [11C]raclopride. First, they describe a theory showing how high-frequency variations in synaptic dopamine leads to lower frequency changes in extracellular levels, which should affect raclopride binding. The theory is then empirically tested using a genetically engineered mouse where DA levels can be manipulated, using FSCV which can record sub-second changes of ESC DA concentrations, and small animal PET to measure raclopride displacement. By decomposing the FSCV signal using wavelet methodology, both low and high-frequency variations could be detected, and the low frequency signal was shown to be correlated to high frequency transient, and affect [11C]raclopride binding. Finally, the method was evaluated in human subjects, where both acute and delayed changes in DA levels in

response to a palatable milkshake could be detected.

This is a very original and exciting study, which makes powerful use of a combination of invasive DA measures and PET in an animal model to validate a new method. The preclinical results are striking, and the application to humans is promising. Although further validation is needed in human subjects, this work may change our ways of designing molecular imaging studies attempting to detect levels of endogenous neurotransmitter levels, and open up new avenues of research into the pathophysiology of a wide range of CNS disorders. The paper should be of high interest to the broad readership of Nature Communications.

Main comments:

There have been previous attempts to obtain data on DA levels, also including methods with high time resolution i.e. the lp-nt PET approach which is also based on bolus + constant infusion. See eg. Morris, J Vis Exp 2013, Cosgrove et al, J Neurosci 2014. Please explain how the proposed method is different, and if/why the authors think their method represents an improvement. The Yale method should be mentioned both in the intro and in the first part of the discussion, where the authors describe previous imaging paradigms of endogenous neurotransmitter measurements. This also applies to the very first sentence in the abstract.

The authors state that rDA cannot be compared between regions, since it calculated as a ratio to absolute local binding, but that subjects can be compared. Wouldn't the relative nature of rDA be a problem here? I.e. what exactly is it that can be compared? They authors state that "within a region rDA is proportional to the amplitude of DA release and can therefore be compared between subjects" (page 15 line 404-405) - however we know that radioligand delivery and possibly non-specific binding (non-displaceable binding) varies between individuals. Could binding data from a compartment showing only non-specific binding (i.e. cerebellum for D2/D3) be incorporated somehow to provide more quantitative information on absolute levels of binding, allowing for comparisons between regions and subjects?

The high temporal resolution (5 mins for human studies) means that there are low counts within every given time period of analysis, particularly towards towards the end of the examination due to radioactive decay - and particularly in low binding regions such as the cortex. This constitutes a fundamental statistical problem affecting signal-to-noise ratio - and is the main reason for why information is usually integrated over the entire PET measurement for traditional quantification. Please comment on the implications.

With regard to extrastriatal regions specifically, the authors state that the relative signal "makes the method more sensitive for the detection of variations in regions with low absolute binding". (Page 15, line 424-425). Another way to see it is that differences within a signal which has a low signal to noise ratio would be expected to be even more unreliable. It is not clear how the data supports the statement "[11C]raclopride can confidently be used as a valuable tracer for the detection of DA release even in extrastriatal regions." (Page 15-16, line 428-429). The authors write that they observed variations in the [11C]raclopride signal in extrastriatal regions (page 13, line 348-349), but the data is not shown. Usually, the level of specific binding relative to non-displaceable (BP_{nd}) in cortical regions is around 0.1 leaving little room for detecting true changes in binding.

The authors searched the whole brain for statistical differences. In addition, for each spatial unit, there are multiple time intervals being compared. How many comparisons were made, in total? The authors identify four regions with early changes, and three with late changes. What is the risk for a type I error? Perhaps a more restricted research volume would improve the statistical approach.

Minor comments:

The paper by Koepp et al (1998) is not the preferred reference for showing effects of endogenous DA on raclopride binding - the authors of this report later published a commendable re-analysis of the data showing that head movement constituted a significant confounder. Whereas the original Nature paper has been cited 648 times according to Web of Science, this much more accurate analysis was only cited 75 times! With regard to the fundamental concept of DA levels affecting binding, any of the early pharmacological challenges would be preferred (and I also suggest these paradigms are discussed first, before moving on to the more debated issue of task-related changes)

I am not sure about this statement: "contribution of noise is the same in each measurement" (page 14-15, line 396-397). The very problem with noise is that is stochastic by nature.

In figure 7 B, it seems that rDA was reduced after activation for one mouse, what was the reason?

Reviewer #4 (Remarks to the Author):

This report describes an ambitious study that seeks to validate detection of rapid extracellular dopamine (DA) transients using PET imaging of labelled raclopride. The findings are very promising. Concerns arise, however, from the basic premise of the study, and a seeming lack of familiarity with relevant literature that would increase the relevance of the experimental findings. The presentation of the modeling studies is particularly problematic, and unlikely to be accessible to most readers. Presentation of the rationale is handled much more clearly and accurately in the Discussion, but this clarity needs to come much earlier when the model and initial results are described.

li. 100. The notion of a defined synaptic volume is not entirely accurate here, given that the majority of DA release sites lack a postsynaptic specialization (Descarries et al 1996). There must also be a typo in li. 105, as the active lifetime of DA in the extracellular space is maximally 100 ms (not seconds) according to cited ref. 17.

101-102. Synaptic "leakage" is a non-standard term. If the authors mean "spillover", this should be used; the rationale for assigning a fractional number to this term (which is governed simply by diffusion) also needs explanation.

109-112. The claims here are not well justified. Extracellular DA concentration is proportional to release, but not directly proportional, given that Michaelis-Menten uptake is a non-linear process. Also unsupported is the statement that, "net changes of extracellular DA concentrations occur at time scales of minutes to seconds and the amplitude of these changes is proportional to extracellular DA concentration...". First, the argument is circular as written, but a more serious concern is the statement that changes in extracellular dopamine might persist for minutes has no basis in fact. Whether in vivo or in ex vivo slices, evoked DA release from coherent stimulation of DA axons leads to evoked release transients that return to baseline within seconds from local uptake – as seen in Supplemental Figure 1. The value of this figure is not clear as it does not seem to support the authors' contention of increases lasting minutes; compounding this inconsistency is that the y axis is in arbitrary units rather than a calculated concentration.

125-128. The model in Figure 1D is inaccurate and misleading, as is the associated text. There are no DA cells in the striatum – rather release is from presynaptic boutons and occasional synapses. Uptake of released DA is into DA axons, not cells. The inaccuracy of the premise underlying the model, call the validity of calculations from the model into question.

li. 151-163. The notion that the overall synaptic space is smaller than ECS volume is accurate.

However, a more relevant reason for focusing on ECS for raclopride displacement is that the vast majority of D2 receptors are extrasynaptic, with less than 10% of DA D2 receptors associated with symmetric synapses (Yung et al 1995). This fundamental characteristic is not discussed, but should be. The authors then go on to say that DA is highest in concentration in a synapse, but this does not matter because the raclopride signal is only from the ECS. Given that DA receptors are predominantly extracellular, this makes an argument for the raclopride signal (reflecting displacement of binding by DA) as originating in the ECS. But instead, they argue that even though the highest DA concentrations, and necessarily the greatest raclopride displacement, will be in synapses, raclopride is primarily extrasynaptic so the signal will be from there. This is not convincing. Instead, there will be little synaptic contribution because 1) there are few postsynaptic specializations for DA release sites, and 2) DA receptors are not found there even when there are. Again, the conceptual underpinnings of the model call the model's calculations into question.

Figure 2. The data presented support activation of DAergic pathways with chemogenetic stimulation. However, some explanation for the rotational behavior should be given, as well as some comment on the greater enhancement of clockwise rotations, which seems odd as mouse dominant hemisphere is left as in most humans. An explanation is offered in the Supplemental Results, greater CNO-enhanced DA release in the left vs. right striatum. This should be noted when the data in Figure 2 are presented, along with an indication of why this might be the case e.g., (DAergic laterality, differential DREADD activation, etc.).

255-272. An increase in frequency of DA transients with CNO is consistent with activation of DA neurons; but presumably the target is also expressed in DA axons, enhancing release. It is therefore surprising that an increase in amplitude was not seen. Might this reflect D2 autoreceptor activation for homeostatic regulation?

303-304; 324-327. The frequencies examined were 0.5 and 0.01 Hz; it is therefore unclear how either of these could be considered "high" frequency. "Higher" but not "high" might be appropriate.

Figure 4. Why does the current envelope decrease with CNO? Also, how the purity of a DA signal was determined is not indicated; what is the evidence that DOPAC, an abundant DA metabolite did not contribute to the current transients detected?

Figure 7. Why does RDA/RDA baseline increase over time without CNO?

318-322. The text states that the highest increases in temporal variation in the raclopride PET imaging was in the left ventral striatum, but "highest" compared to what? Was this vs. right ventral striatum, or left or right dorsal striatum? Was dorsal striatum even monitored? If so, were changes significant? Was there laterality?

339. The composition of the "tasteless" solution should be indicated in the text, particularly whether this was nutritive or not. This is also omitted from the methods section. As presented (346-353), the use of "tasteless" as the control solution implies that the difference is between "taste" and lack of taste. However, it is well known that insulin levels rise after ingestion of a nutritive solution, and that insulin crossed the blood brain barrier where it activates insulin receptor signaling and can amplify striatal DA release (e.g., Stouffer et al 2015; Woods et al 2016). This is likely to be a contributing factor to both the early and later phase increases in PET response. In this light, the most exciting aspect of the present studies is the differential timecourse of the PET response in ventral vs. dorsal striatum, although the explanation given omits the likely contribution of nutritive signaling.

Minor

78. Should be "of" not "for".

121. No need to redefine ECS here.

General. Figure should be referenced in the text individually in parentheses, rather as often presented here, at the end of a list of statistical results for a given experiment. It is difficult to find the corresponding figure that illustrates some sections.

338. Human subjects were monitored, not measured.

We would like to thank all the reviewers for their constructive comments. Based on these, we have conducted new experiments and performed additional analyses to address the reviewers' concerns. We are pleased that these all support our overall interpretation and we believe the manuscript is now improved from the original submission. The specific points are addressed below.

Reviewers' comments:

Reviewer #1 (Remarks to the Author):

The manuscript by Lippert et al presents an innovative method of analyzing PET data that purports to harness the dynamic changes in the PET signal to provide information about underlying fluctuations in dopamine release. To support their claims the authors combine modelling with fast-scan cyclic voltammetry in mice and PET experiments in mice and humans. I have direct experience with voltammetry but not with PET and I have limited modeling experience. Thus, I feel better equipped to judge some of their claims than others. However, despite my limited experience I found the report easy to read and gain a decent understanding of the theory, the experiments, and the potential significance. The basic idea, as far as I can ascertain, is that second-to-second fluctuations in dopamine release are well-correlated with minute-to-minute fluctuations and this correlation allows the dynamic PET signal, analyzed in this novel way, to act as a proxy for the changes in release that are usually measured with invasive techniques such as voltammetry.

This paper and the technique it introduces could be potentially of great interest to many and may shift the way PET data is dealt with, analyzed, and interpreted in the future. I do have a number of questions and suggestions that should be taken into account by the authors and editor.

Major comment(s)

My main point is that while I am fairly well convinced that this new method of analysis may pick up differences in regional dopamine function, as the voltammetry data is taken from a single type of manipulation (CNO-evoked transients), I am wary of the interpretation becoming too broad. For example, I do not believe enough has been done to firmly state that the difference reported in the PET signal is due to an increase in transient number, or even dopamine release events.

For example, the manipulation that the authors use may produce a certain set of results (e.g. the correlations between transient number, and high and low frequency wavelet power) that might not exist if a different method of altering dopamine function had been used. This is pertinent because of the possibility that the slow ramping of the signals in this model could potentially lead to artifacts in the wavelet transformation at low powers. The study would have been significantly strengthened if the authors had managed to increase (and even suppress) the number of dopamine transients in a more intermittent, on-off fashion, for example, by using optogenetics or a pharmacological manipulation with a short half-life.

We appreciate the reviewer's point that this is an important consideration. Nonetheless, we are confident that the method will be widely applicable for a number of reasons. First, the application of the method to the human data and the detections of food-induced DA release support the argument that the activation pattern we see in the experimental data is not resulting solely from the model system used.

Second, as suggested by the reviewer, we performed additional continuous FSCV recordings in the ventral striatum of two wild-type mice. Here, instead of using activation of a DREADD receptor with CNO, we now induced DA release in separate one-minute time intervals by electrical stimulation of the VTA at a rate of either 5 or 10 transients per minute followed by a resting time interval of either 4 minutes (mouse 1) or 9 minutes (mouse 2). In line with the data recorded in chemogenetically activated mice, the wavelet power at 0.5 Hz correlated with transient rates and - crucially - also correlated significantly with the power at 0.01 Hz. These data support our message that transient DA release not only induces second-by-second but also minute-by-minute fluctuations in extracellular DA levels. These data are shown in the new Figure 6 and discussed in line 257 and lines 306-312 and 328-331. Experimental procedure is described in lines 610-615.

Unfortunately, we are not able to provide additional [11C]raclopride PET data because the usage of [11C]raclopride requires an onsite cyclotron due to the short half life of 11C (20 minutes). After completion of this study our cyclotron was shut down and will only be replaced in the future (>2 years).

However, we can provide data from two published PET studies that support our claim that DA release induces minute-by-minute fluctuations in [11C]raclopride PET data rather long-lasting decreases in the signal. Both studies applied a bolus plus constant infusion protocol, which is a premise for our method, to inject [11C]raclopride into human subjects. Breier et al. (1997) analysed DA release after amphetamine administration in schizophrenic patients. The amphetamine-induced DA release was so strong that it reduced the net signal in the striatum. However, their exemplary data of one patient show that the temporal fluctuations in the striatal data are increased in the post-injection period. From our theory we would interpret these temporal variations (rDA) as repeated phasic DA release. This would mean that amphetamine not only induces a single huge release of DA at the time of injection but also continuing phasic DA release. Such patterns of increased phasic release following amphetamine administration are also seen in rodents using voltammetry (Daberkow et al. 2013).

In the second study a group in Yale analysed smoking-induced DA release by letting subjects smoke cigarettes during $[^{11}\text{C}]$ raclopride PET data acquisition (Morris et al., 2013.; Cosgrove et al. 2014). The published time activity curve recorded from a voxel in the striatum again indicates a clear increase in temporal fluctuations at the start of smoking rather than a decrease in the signal (Morris et al. J Vis Exp, 2013):

Smoking started at 45 minutes. Correction for head motion was applied to the data such that an artefact by head motion can be excluded.

If desired by the reviewers/editors we could add this information to the Supplemental Information.

Another related issue involves the time course of transients. Within the striatum, duration of transients varies significantly due to differences in DAT expression. Other extrastriatal brain regions, have vastly different levels of DAT, which will affect transient duration. In addition, different release patterns of dopamine may also exist depending on the firing of projecting cells and terminal mechanisms. It may be significant that I am not aware of spontaneous dopamine transients being recorded with voltammetry in any extrastriatal region. What implications does this have for the authors' PET findings in extrastriatal regions.

Thanks for raising this issue. The reviewer is correct stating that DAT expression varies between regions. But there are indications that it also varies on a subcellular level within regions. FSCV data recorded in essentially all regions show contributions from fast domains and slow domains. DA is removed from the fast domain within seconds reflecting high density of DATs and from the slow domain within a minute reflecting low density of DATs. Heterogeneity of DAT expression on subcellular levels could be responsible for the effect we observe. Recent model calculations by Kaya et al. (2018)⁴⁰ showed that when DATs are expressed heterogeneously, some of the released DA diffuses into subregions with low DAT expression and accordingly longer lifetime. Slow removal of DA is able to modulate the PET signal.

In extrastriatal regions the densities of DA synapses, receptors, and transporters are much lower than in the striatum (e.g. Sasaki et al., J Nucl Med, 2012). An implanted FSCV electrode in the cortex therefore effectively receives input from fewer synapses than in the striatum which tremendously reduces the amplitude of the FSCV signal and makes it more difficult to detect. However, the overall lower density of the DATs in extrastriatal regions leads to an overall longer lifetime there. Or in other words: there is a much stronger contribution from the slow domain. This means that although less DA is released in extrastriatal regions, nearly all of the released DA is removed at longer time scales and thereby detectable by [11C]raclopride PET. This could explain the clear extrastriatal responses we observe in the human data. Extrastriatal FSCV recordings of evoked DA release confirm this picture (Garris et al. 1993, see revised Figure 2). We added this aspect to the manuscript:

Lines 447-452: "Moreover, minute scale removal rates are often present in FSCV transient recordings – sometimes referred to as "hang-up". As discussed by A. Michael and colleagues "Evoked responses ... with hang-up are absolutely commonplace." A potential mechanism that could explain the slow minute scale component in the FSCV data could be the heterogeneous subcellular distribution of DA transporters (DATs). A fraction of the released DA diffuses into subcellular regions with low DAT expression and accordingly slow removal rates (Figure 2)."

Lines 463-468: "Another property of the DA system promotes the detection of DA release in terms of minute-by-minute variations in extrastriatal regions: although less DA is released in extrastriatal regions due the lower density of DA synapses, the major fraction of released

DA is removed at a minute time scale due to the lower density of DATs. This means that the amplitudes of release-induced minute-by-minute variations in extracellular DA concentrations are of the same order of magnitude in extrastriatal regions as in the striatum despite the difference in total amount of released DA (revised Figure 2)."

Minor

1. To what extent will PET experiments that have already been conducted be able to benefit from this new analysis method? The authors mention that between-region analysis is uninformative and so experimental design and data collection will need to have been conducted in a certain way. This is not a criticism of the study but something that the authors may wish to comment on in a little more detail.

We thank the reviewer for touching on this point. The only requirement to apply our method to [11C]raclopride (or other receptor ligand) PET data is that the PET tracer is in steady state, i.e. delivered by a bolus plus constant infusion injection. This method of tracer injection was performed at the NIH (Breier et al., 1997) and in Yale (Normandin et al., 2012) before (see previous answer). We are very curious how a reanalysis of these data with our method would look like. The examples published by Breier et al. and Normandin et al. already show an obvious increase in temporal fluctuations of the PET signal after induction of DA release in human subjects. It would be interesting to see if our method is able to identify additional responses in their data. In our human data after food intake we get significant results with our method although there is no net reduction in the [11C]raclopride signal.

We added a paragraph to the discussion about the requirements to inspire other labs to reanalyse their data (lines 511-519).

2. Equally, the authors' claims about the correlation between low and high frequency wavelet power informing the long-running discussion about phasic vs. tonic dopamine is intriguing and attractive. They could think about expanding this a little.

We thank the reviewer for this encouraging comment. There is indeed controversy about the functions of phasic and tonic dopamine release. Therefore we didn't want to elaborate too much on this subject as our data are not sufficient to resolve this argument but rather can stimulate further discussion in the field. We culminated our discussion with the sentence: "If minute-by-minute levels are just a consequence of second-by-second release, evoked functions are presumably related to each other." We here cite the paper of Hamid et al. (2016), in which they relate functions to tonic DA levels (assessed by microdialysis) and phasic DA levels (assessed by FSCV) in the nucleus accumbens acquired simultaneously in rats while performing a behavioural task.

In order to clearly make the connection to tonic and phasic DA we added: "In other words: if phasic DA release effectively determines the tonic DA level, the functional consequences of both might be related. Further discussion of this topic can be found in a recent article by J.D. Berke" (lines 507-509)

Additional experimental data beyond the scope of this manuscript are required to bring more clarity.

3. More information should be provided on the mouse line used – e.g. body weights, have they been previously published, any other phenotypic observations? In addition, it should be noted that all DAT-expressing neurons will be affected in this model, not just midbrain dopamine neurons.

While the specific model ($hM3D_{Gq}^{DAT}$) has not been previously studied, other groups have used viral expression of $hM3D_{Gq}$ to successfully stimulate locomotor behaviour in DAT-Cre animals (Wang et al. Neuroscience Bulletin, 2013). Furthermore, our group has used additional Cre systems in combination with the $hM3D_{Gq}$ mouse to stimulate responses (Steculorum et al., reference 54). We have now added the body weight information to the text (line 233-235: “At baseline, wild-type and transgenic animals displayed no differences in body weight ($hM3D_{Gq}^{WT}$: 22.91g +/- 1.23 vs $hM3D_{Gq}^{DAT}$: 22.21g +/- 1.61, $p=0.19$) nor did they display any differences in normal behavior.”) to show that no basal bodyweight phenotype exists in these animals. Note, that the data in Figure 2 indicates that $hM3D_{Gq}^{DAT}$ animals receiving only saline display no abnormal locomotor phenotype.

We agree that all dopamine neurons would be affected in this model, and the text states that the $hM3D_{Gq}$ would be targeted to all DAT expressing neurons (line 230). Despite this global expression in all DAT neurons, a primary effect we see after CNO administration is increased locomotor activity, pointing to the predominant effect on activating the midbrain dopamine neurons involved in motor movement. This readout was also used to confirm that we do in fact have activation of the dopamine system before moving forward with the more intense methodological work. There are some suggestions that additional dopaminergic populations (e.g. the zona incerta) may also send projections to regions in the striatum. However, it is well established that the midbrain dopamine neurons are the primary source of dopamine for striatal regions and would provide the major fraction of dopamine release detected in our study.

4. Showing the stimulated release data would be useful addition – is it the same between genotypes?

In the PET and FSCV experiments, all experiments were performed in DAT-Cre tg/wt; $hM3D_{Gq}$ ($hM3D_{Gq}^{DAT}$) animals. Therefore the differences noted between the groups would only be due to the stimulation of dopamine release after treatment with CNO. We did not assess stimulated release in non-transgenic versus transgenic animals to ensure that all experiments were performed in animals with the same genetic background ($hM3D_{Gq}^{DAT}$). However, recorded stimulus-response curves of the $hM3D_{Gq}^{DAT}$ mice do not appear to be different from published recordings in wild-type mice.

5. Could the authors make it easier to see which trace is which in panel 3B - the overlays with the standard error are difficult to interpret. One possibility would be to show each trace side by side with its standard error and then an overlay with just the averages?

Thank you for pointing this out. We have adjusted the figure accordingly.

6. The authors state that “[Despite] minor differences in spontaneous baseline activity apparent in the first 5 minutes of baseline recording...”. This difference actually seems quite significant. What’s the interpretation? Also, were these data included in their ANOVA?

Given the low numbers of animals needed to determine the effect of CNO on dopamine release, it is not unusual that we might see much more variance in the data as the animals are adjusted to baseline. Of note, is that the 5 minutes prior to CNO injection (see the Figure below showing minute 1-10 of Figure 4C enlarged) shows no difference between animals, and thus provides a stable baseline with no transients prior to injection.

For the ANOVA of the stimulation, we performed the analysis from the time of injection to the end of the first hour, and then the second hour was performed individually. We did not include the baseline in this calculation. We are of the opinion that the ANOVA should be performed on the period immediately following the injection and the baseline should be excluded from this analysis. However, if we do include the baseline data in this calculation, it only further increases the significant effect of time and treatment ($p_{\text{time}}=0.0019$, $p_{\text{treatment}}=0.0084$). Therefore, the effect is present with or without the inclusion of the baseline data in the analysis.

7. In the individual mouse trace (Fig 5A, top) comparing transient rate and power in the 0.5 Hz band, although the signals tend to be correlated, there are times at which these signals deviate. Do the authors have any hypothesis as to what causes these deviations? Could they be related, for example, to a change in transient properties (e.g. transients with double-peaks or a longer duration)?

Counting of transient rates per minute was performed using a template representing a typical transient. A structure was counted as a transient when its correlation with the template was higher than 0.86. This threshold definitely leads to both, false positive and false negative results. As suggested by the reviewer, changes in transient properties will reduce the detectability of transients by this method.

The big advantage of the wavelet transform is the fact that it is a well defined formalism without any adjustable parameters, i.e. a wavelet transform performed by lab A provides identical results when performed by lab B. There is no subjective component in this analysis. Therefore we think the power of the 0.5 Hz band of the wavelet transform is the better measure of phasic dopaminergic activity. At least a double peak would still increase the wavelet power while it could hamper detection with the template. However, as far as we know, this is the first time this type of analysis was applied to continuous FSCV data and this aspect certainly requires further investigation.

8. In Fig 7 it would be good to show where in the brain these data are taken from, for example, with a simple schematic.

Following the suggestion of the reviewer we added a schematic to Figure 9 (and also Supplemental Figure 3) indicating the analysed locations.

9. In Fig 8, the caption should more clearly state what is going on in each panel. It describes everything quite well but is difficult to quickly find what is in panel C, for example.

We reorganised the caption of the revised Figure 10 to increase clarity.

Reviewer #2 (Remarks to the Author):

The manuscript by Backes and colleagues outlines a novel method of analyzing PET signals to greatly increase the temporal resolution of the technique. They propose a mathematical model to relate slow changes in D2 receptor-bound raclopride levels to fast increases in extracellular dopamine due to synaptic activity. Fast scan cyclic voltammetry (FSCV) in anesthetized mice was used to validate the model. The continuous FSCV methods used to monitor rapid spontaneous synchronous dopamine release events (transients) appear to be appropriate and rigorous, and the analysis of the FSCV data is both elegant and novel. Chemogenetic activation of dopamine neurons with CNO is used to increase extracellular dopamine, and both FSCV and the new PET method are able to measure the changes. CNO has a fairly rapid onset, on the order of several minutes, but has a slow offset (30 min-2 hours depending on dose), and therefore may not be the best choice for testing the limits of the PET model. Repeated electrical stimulations of VTA neurons for 1 second or less might provide a better challenge to test the authors' claim that minute-by-minute fluctuations of the PET signal relate directly to subsecond dopamine release events.

Thank you for this helpful suggestion. This comment is nearly identical to one raised by reviewer 1 (point 1). Therefore we repeat our reply from above.

As suggested by the reviewer, we performed additional continuous FSCV recordings in the ventral striatum of two wild-type mice. Here, instead of using activation of a DREADD receptor with CNO, we now induced DA release in separate one-minute time intervals by electrical stimulation of the VTA at a rate of either 5 or 10 transients per minute followed by a resting time interval of either 4 minutes (mouse 1) or 9 minutes (mouse 2). In line with the data recorded in chemogenetically activated mice, the wavelet power at 0.5 Hz correlated with transient rates and - crucially - also correlated significantly with the power at 0.01 Hz. These data support our message that transient DA release not only induces second-by-second but also minute-by-minute fluctuations in extracellular DA levels. These data are shown in the new Figure 6 and discussed in line 257 and lines 306-312 and 328-331. Experimental procedure is described in lines 610-615.

Unfortunately, we are not able to provide additional [11C]raclopride PET data because the usage of [11C]raclopride requires an onsite cyclotron due to the short half life of 11C (20 minutes) and our cyclotron is currently replaced and will be out of function for a long time.

However, we can provide data from two published PET studies that support our claim that DA release induces minute-by-minute fluctuations in [11C]raclopride PET data rather long-lasting decreases in the signal. Both studies applied a bolus plus constant infusion protocol, which is a premise for our method, to inject [11C]raclopride into human subjects. Breier et al. (1997) analysed DA release after amphetamine administration. However, their data also show that the temporal fluctuations in the striatal data are increased in the post-injection period. From our theory we would interpret these temporal variations (rDA) repeated phasic DA release. This would mean that amphetamine not only induces a single huge release of DA at the time of injection but also continuing phasic DA release. Such patterns of increased phasic release following amphetamine administration are also seen in rodents using voltammetry (Daberkow et al. 2013).

In the second study a group in Yale analysed smoking-induced DA release by letting subjects smoke cigarettes during [11C]raclopride PET data acquisition (Morris et al., 2013.; Cosgrove et al. 2014). The published time activity curve recorded from a voxel in the

striatum here also indicates a clear increase in temporal fluctuations at the start of smoking rather than a decrease in the signal (Morris et al. J Vis Exp, 2013):

Smoking started at 45 minutes. Correction for head motion was applied to the data such that an artefact by head motion can be excluded.

If desired by the reviewers/editors we could add this information to the Supplemental Information.

Also, the manuscript needs to include an in-depth discussion of the work of Kishida and Montague describing FSCV measurements of real-time spontaneous dopamine and serotonin signals in human brain during decision-making tasks (Kishida et al., 2016; Moran et al., 2018), and the relationship between rapid dopamine signaling and BOLD responses in human subjects (Lohrenz et al., 2016).

We included the suggested references to point out that FSCV recordings have been performed to directly assess task-related neurotransmitter release in the human brain (line 363-364). We see the scientific impact of the research highlighted here by the reviewer. However, given that the main focus of our work is methodological and not discussed in the context of functional consequences, an in-depth discussion of this work would be out of the scope of our work.

Reviewer #3 (Remarks to the Author):

In this paper, Lippert and colleagues evaluate a novel method for measuring changes in synaptic dopamine release using PET and the D2/D3 antagonist radioligand [11C]raclopride. First, they describe a theory showing how high-frequency variations in synaptic dopamine leads to lower frequency changes in extracellular levels, which should affect raclopride binding. The theory is then empirically tested using a genetically engineered mouse where DA levels can be manipulated, using FSCV which can record sub-second changes of ESC DA concentrations, and small animal PET to measure raclopride displacement. By decomposing the FSCV signal using wavelet methodology, both low and high-frequency variations could be detected, and the low frequency signal was shown to be correlated to high frequency transient, and affect

[11C]raclopride binding. Finally, the method was evaluated in human subjects, where both acute and delayed changes in DA levels in response to a palatable milkshake could be detected.

This is a very original and exciting study, which makes powerful use of a combination of invasive DA measures and PET in an animal model to validate a new method. The preclinical results are striking, and the application to humans is promising. Although further validation is needed in human subjects, this work may change our ways of designing molecular imaging studies attempting to detect levels of endogenous neurotransmitter levels, and open up new avenues of research into the pathophysiology of a wide range of CNS disorders. The paper should be of high interest to the broad readership of Nature Communications.

Main comments:

There have been previous attempts to obtain data on DA levels, also including methods with high time resolution i.e. the lp-nt PET approach which is also based on bolus + constant infusion. See eg. Morris, J Vis Exp 2013, Cosgrove et al, J Neurosci 2014. Please explain how the proposed method is different, and if/why the authors think their method represents an improvement. The Yale method should be mentioned both in the intro and in the first part of the discussion, where the authors describe previous imaging paradigms of endogenous neurotransmitter measurements. This also applies to the very first sentence in the abstract.

We thank the reviewer for mentioning the lp-ntPET method, which of course should be discussed in our manuscript. The Yale method belongs to the second type of approaches in our discussion: "In the second approach, the kinetic model for the PET tracer was extended in order to account for dynamic changes of endogenous DA concentrations." The fitting equation is identical to that introduced by Alpert et al. (2003, Eq. 6 with $\alpha=0$ and $\beta=0$ is identical to the operational equation in Normandin et al. 2012 and Cosgrove et al. 2014) but the functional expression for the time-dependent DA release differs (Normandin et al. 2012). As in the other approaches, the Yale method will only detect DA release if there is a significant deviation in the TAC such that taking into account a DA function $h(t)$ significantly improves the fit of the data. Thus, our comment in the discussion: "...both approaches... require robust and long-lasting (>minute timescale) increases of DA concentrations to significantly reduce net regional tracer binding, in order to produce detectable DA release events." also applies to the Yale method.

In contrast, our approach is not model-based and requires no fitting of the data. The parameter r_{DA} is directly calculated from the PET data as detailed in Equation 10. We only use the well-established compartment model to explain how temporal variations in extracellular DA concentrations can induce temporal fluctuations in the [11C]raclopride PET data. Neither in our human data nor in the mouse PET data did we observe significant reductions of the PET signal (i.e. the Yale method would not detect DA release here; see Figure 10C) but we observed significant increase in temporal fluctuations. As we write in the discussion: "FSCV data recorded in situ in mice indicate that phasic DA release induces minute-by-minute fluctuations rather than minute-lasting elevations of DA concentrations. This implies that temporal variations in the PET signal are far more sensitive for the detection of DA release events than the net time activity curve."

However, the Yale group also used a bolus plus constant infusion method for tracer injection. Therefore our method can be readily applied to their data.

As suggested by the reviewer we added references to the Yale method to the Introduction and Discussion.

The authors state that rDA cannot be compared between regions, since it is calculated as a ratio to absolute local binding, but that subjects can be compared. Wouldn't the relative nature of rDA be a problem here? I.e. what exactly is it that can be compared? The authors state that "within a region rDA is proportional to the amplitude of DA release and can therefore be compared between subjects" (page 15 line 404-405) - however we know that radioligand delivery and possibly non-specific binding (non-displaceable binding) varies between individuals. Could binding data from a compartment showing only non-specific binding (i.e. cerebellum for D2/D3) be incorporated somehow to provide more quantitative information on absolute levels of binding, allowing for comparisons between regions and subjects?

The absolute value of rDA (the ratio of temporal variations to absolute [11C]raclopride signal) depends on several factors: (1) the absolute raclopride signal (because it is normalized to it), (2) the amount of released DA, (3) the fraction of released DA which is removed at a minute time scale, (4) noise resulting from the imperfection of the method (such as stochastic nature of radioactive decay, <100% efficiency of detecting decay events, etc.). The contribution of (4) causes the steady increase of rDA with time due to the short half life of [11C]raclopride as we point out in the discussion. Ensuring stable conditions (same PET scanner, comparable injected tracer activity, etc.) one can be confident that the contribution of (4) is about the same for each subject. (1), (2), and (3) are region-specific and can also vary between subjects. For example in extrastriatal regions the contribution of (1) is lower than in the striatum because of the lower density of D2 receptors (increasing rDA), the contribution of (2) is also lower because of the lower density of synapses (decreasing rDA), and the contribution of (3) is higher because of the lower density of DA transporters that remove the DA (increasing rDA). We think that this is the reason why DA releases in extrastriatal regions cause changes in rDA with similar amplitudes as in the striatum (see revised Figure 2): although less DA is released and reaches the extracellular space, the predominant part is removed by the DATs at a much slower rate than in the striatum. This view is supported by simultaneous FSCV recordings from striatal and extrastriatal regions from the Wightman group (Garris et al., see revised Figure 2). These data show that the amplitude of the DA fraction that is removed slowly and that causes the variations of rDA is the same in the striatum and in the medial prefrontal cortex: combining the data of CP and MPFC in the same plot shows that the tail representing slowly removed DA (data at 20 seconds) is identical in both regions (see revised Figure 2). This argumentation supports our statement that rDA indeed represents DA release rates but cannot be absolutely compared between regions.

In our human data (currently in revision in Cell Metabolism, the manuscript was provided to the reviewers) we find high similarity of rDA within a region between subjects (see Figure 7 in the supporting manuscript). We could, for example, show that the absolute change of rDA in the insular cortex during intake of milkshake in comparison to a tasteless solution was proportional to the subjective "wanting" scale of the subjects at a high significance level ($p=0.0001$) and with a high linear correlation coefficient ($r=0.94$). Although this isn't a "hard proof" it indicates that rDA within a region can be compared between subjects. Taking into account the cerebellum as a reference region would not necessarily increase comparability it would rather introduce another uncertain

parameter: who can really guarantee that the cerebellum represents whole-brain unspecific binding?

The high temporal resolution (5 mins for human studies) means that there are low counts within every given time period of analysis, particularly towards the end of the examination due to radioactive decay - and particularly in low binding regions such as the cortex. This constitutes a fundamental statistical problem affecting signal-to-noise ratio - and is the main reason for why information is usually integrated over the entire PET measurement for traditional quantification. Please comment on the implications.

In order to gain statistical confidence, rDA is calculated from the temporal variations in a box of neighbouring voxels (125 in the human data). Within such a box in the cortex we count in a 5-minute interval at the end of the scan time ~30000 decay events. As discussed in the manuscript (line 428-430), radioactive decay and the resulting decrease of signal-to-noise ratio cause the steady increase of rDA with time which can be observed in human and in mouse data. In spite of this we are still able to observe significant stimulation-related regional increases of rDA in mouse and human data. As discussed above, the comparability of these data between subjects gives us some confidence that we are able to detect rDA changes at a time resolution of 5 minutes.

The emphasis of our method was to try a completely different approach to gain spatial as well as temporal information on DA release from the PET data. The starting point was the observation of temporal fluctuations in the human [11C]raclopride data at reasonable locations and reasonable time points that could not be explained by integration methods. Integration provides robust read-outs at the expense of losing dynamic information.

With regard to extrastriatal regions specifically, the authors state that the relative signal "makes the method more sensitive for the detection of variations in regions with low absolute binding". (Page 15, line 424-425). Another way to see it is that differences within a signal which has a low signal to noise ratio would be expected to be even more unreliable. It is not clear how the data supports the statement "[11C]raclopride can confidently be used as a valuable tracer for the detection of DA release even in extrastriatal regions." (Page 15-16, line 428-429). The authors write that they observed variations in the [11C]raclopride signal in extrastriatal regions (page 13, line 348-349), but the data is not shown. Usually, the level of specific binding relative to non-displaceable (BP_{nd}) in cortical regions is around 0.1 leaving little room for detecting true changes in binding.

In revised Figure 10 (Figure 8 in the original manuscript) we show variations in the extrastriatal [11C]raclopride signal in the midbrain and brainstem (1-NTS, 2-SN/VTA, 5-Pons, 6-VPM). The human PET data is shown here to demonstrate the power of the method to detect DA release in humans. These data are discussed in full length in a second manuscript which was provided to the reviewers as supporting material. In the human paper we report additional extrastriatal regions with significant food-induced increase of temporal variations in the [11C]raclopride signal such as the prefrontal cortex and the insular cortex.

We agree with the reviewer that the lower signal and thereby lower value for normalization alone cannot explain why the observed amplitudes of variation are similar in extrastriatal and striatal regions. Although it explains part of it: as we show with model calculations (Figure 1F) we expect that the amplitude of [11C]raclopride variations in

extrastriatal regions is only a factor of ~5 lower than in the striatum although the density of D2 receptors is a factor of >10 lower. We think that another factor (already discussed above) increases the sensitivity for [11C]raclopride variations to DA release: due to the lower density of DA transporters in extrastriatal regions the major fraction of released DA is removed slowly from the extracellular space while in the striatum the major fraction is removed within seconds and only a minor fraction is removed at a minute time scale. This aspect was not discussed in the original version of the manuscript but is included in the revised version (line 447-452 and revised Figure 2).

The authors searched the whole brain for statistical differences. In addition, for each spatial unit, there are multiple time intervals being compared. How many comparisons were made, in total? The authors identify four regions with early changes, and three with late changes. What is the risk for a type I error? Perhaps a more restricted research volume would improve the statistical approach.

We identified two time intervals of interest from a whole brain analysis by simply counting the number of significant voxels in the whole brain for each time interval (see Figure 2 in Thanarajah et al. provided as material). Then we performed paired t-tests for these two time intervals. Apart from the seven regions shown here we found additional regions that are discussed in the human manuscript. We applied corrections for multiple comparisons on cluster level following Friston et al. (Hum Brain Mapp, 1994). All seven regions remained significant after correction. We added the corresponding p-values to the Results section (line 377-388) and referenced the method for correction in the Methods section (lines 781-782).

Restricting the search volume would not change the data within that volume. For example if we identified a spot within the striatum with a significant uncorrected increase of rDA (pairwise t-test) it could be that it would not survive correction for multiple comparison if we take into account the whole brain. If instead we would restrict the search to the striatum, the identical data would show a significant increase even after correction for multiple comparisons. But the true chance of type I error must be identical for both analyses because both are based on the same set of data. This means that the gain of significance by restriction of search volume is artificial and does not reduce the true risk for type I errors.

Minor comments:

The paper by Koeppe et al (1998) is not the preferred reference for showing effects of endogenous DA on raclopride binding - the authors of this report later published a commendable re-analysis of the data showing that head movement constituted a significant confounder. Whereas the original Nature paper has been cited 648 times according to Web of Science, this much more accurate analysis was only cited 75 times! With regard to the fundamental concept of DA levels affecting binding, any of the early pharmacological challenges would be preferred (and I also suggest these paradigms are discussed first, before moving on to the more debated issue of task-related changes)

We thank the reviewer for pointing out that Koeppe et al. (1998) is not the best reference and that the pioneers of the method should be cited here. We included a reference to Dewey

et al. (Synapse, 1993) which is an early report on alterations of [11C]raclopride binding by pharmacological manipulation of the endogenous DA system.

I am not sure about this statement: "contribution of noise is the same in each measurement" (page 14-15, line 396-397). The very problem with noise is that is stochastic by nature.

The noise level is quite stable but single events that cause the noise are stochastic. The signal-to-noise ratio depends on the apparatus and the circumstances (temperature, humidity, pressure, etc.) of the measurement. Repeating a measurement at similar circumstances with the same apparatus leads to the same signal-to-noise ratio. In our human data we see a high regional similarity of rDA between subjects. However, it is essential to ensure that the conditions are the same for each measurement as we did in our study. For example, no new PET setup should be performed between measurements. We added a comment to the discussion (line 518-519).

In figure 7 B, it seems that rDA was reduced after activation for one mouse, what was the reason?

While we don't have a definite answer as to why this occurred, we know from our experience with i.p. catheter placement that occasionally the substance is not injected into the peritoneum and accordingly does not reach the blood stream. However, we have no method to directly detect if this has occurred here and therefore did not exclude this animal from the analysis. On the other hand the difference could represent biological variability. Despite this animal, the overall effect is still evident in the data.

Reviewer #4 (Remarks to the Author):

This report describes an ambitious study that seeks to validate detection of rapid extracellular dopamine (DA) transients using PET imaging of labelled raclopride. The findings are very promising. Concerns arise, however, from the basic premise of the study, and a seeming lack of familiarity with relevant literature that would increase the relevance of the experimental findings. The presentation of the modeling studies is particularly problematic, and unlikely to be accessible to most readers. Presentation of the rationale is handled much more clearly and accurately in the Discussion, but this clarity needs to come much earlier when the model and initial results are described.

Thank you for pointing out the issues with the presentation of the model. We appreciate this as it is important to us that the details are accessible to a wide readership.

li. 100. The notion of a defined synaptic volume is not entirely accurate here, given that the majority of DA release sites lack a postsynaptic specialization (Descarries et al 1996).

One prerequisite of our model is that DA is released from a presynaptic site while the postsynaptic target does not matter. Using Eq. 1 to describe spillover from a synapse after phasic release of DA is well established in literature (e.g. ref 17, Rice and Cragg, 2008). The definite values given in the brackets only reflect "typical" values indicating the order of magnitude. To calculate spillover from a defined synapse one would have to insert the volume of this synapse and the fraction of synaptic spillover for this synapse.

There must also be a typo in li. 105, as the active lifetime of DA in the extracellular space is maximally 100 ms (not seconds) according to cited ref. 17.

We thank the reviewer for pointing out this inaccuracy. It is not a typo but reference 17 is not the best reference here. They give three different rate constants for dopamine removal (0.007/sec, 0.12/sec, and 20/sec) representing average uptake in different (macroscopic) regions (unspecific, SNc, and Striatum). The minute timescale, referred to as “hang-up” or “slow domain”, was observed in FSCV data throughout the striatum and nucleus accumbens and was discussed by A. Michael and colleagues: “Evoked responses ... with hang-up are absolutely commonplace”. We corrected the mistake and added the reference Taylor et al. (2015). We also added an additional Figure (revised Figure 2) to the manuscript which indicates the presence of a fast and a slow domain in extracellular DA concentrations calculated from Equation 1 and measured by FSCV.

101-102. Synaptic “leakage” is a non-standard term. If the authors mean “spillover”, this should be used; the rationale for assigning a fractional number to this term (which is governed simply by diffusion) also needs explanation.

In relation to PET data the term “spillover” is typically used to describe the effect of signal spreading to neighbouring image voxels, i.e. an artefact caused by technical limitations. That was the reason why we used the term “leakage” here to point out that synapses are not “leak proof” and that DA molecules driven by diffusion leak out of the synapses. Following the reviewer’s suggestion we added the term “spillover”.

Please note that the numbers given in line 103 only express the order of magnitude and reflect typical values found in literature. We explicitly use the term l_s in Equation (1) because it strictly applies only to dopamine diffusion in the extracellular space outside the synapse. Within the synapse the narrow space confined by pre- and postsynaptic membranes and the high density of transporters and receptors would require a mathematical description different from Equation (1). However, Equation (1) successfully describes the dopamine concentrations in the extracellular space observed after phasic release. But then not the total amount of released dopamine but rather the fraction that was able to diffuse out of the synapse should be inserted into Equation (1). Here we estimate this fraction to be in the order of 1%. We added this information to the revised manuscript (lines 103-109).

109-112. The claims here are not well justified. Extracellular DA concentration is proportional to release, but not directly proportional, given that Michaelis-Menten uptake is a non-linear process.

The reviewer is correct pointing out that we use the linear approximation here for Michaelis-Menten kinetics. The linear approximation for Michaelis-Menten kinetics is valid if the concentration is much lower than the Michaelis-Menten constant K_m (see e.g. Sokoloff 1974). It is a common approximation to describe DA kinetics in the extracellular space (Rice and Cragg, 2008). Since DA concentration in the extracellular space is determined by diffusion, it decreases as a function of distance from the synapse and the linear approximation is valid in the major part of the extracellular space. If the concentration is $\geq K_m$ the linear estimation overestimates the removal rate, i.e. real lifetime of DA is longer than predicted by the model. However, the fact that Eq. 1 provides realistic

fits to measured transient data (Rice and Cragg, 2008) indicates that the linear approximation appears to be valid.

Also unsupported is the statement that, “net changes of extracellular DA concentrations occur at time scales of minutes to seconds and the amplitude of these changes is proportional to extracellular DA concentration...”. First, the argument is circular as written, but a more serious concern is the statement that changes in extracellular dopamine might persist for minutes has no basis in fact. Whether in vivo or in ex vivo slices, evoked DA release from coherent stimulation of DA axons leads to evoked release transients that return to baseline within seconds from local uptake – as seen in Supplemental Figure 1. The value of this figure is not clear as it does not seem to support the authors’ contention of increases lasting minutes; compounding this inconsistency is that the y axis is in arbitrary units rather than a calculated concentration.

The statement that dopamine persists for minutes is well supported by FSCV data (see publications by Adrian Michael). The observed transients exist of the summed signal from slow and fast domains. The reviewer is correct that the most of the released DA is removed within seconds by the DATs within the fast domains because the amplitude of the contribution from the slow domain is much lower than the amplitude of the contribution from the fast domain. But the contribution from the slow domain(s) is still evident in the data. A potential source for the existence of slow domains was discussed in a recent publication by Kaya et al. (2018)⁴⁰ in terms of a heterogeneous subcellular density of DATs: “In principle, one might expect that the increased population of DATs in dense regions would counterbalance the effect of lower surface area coverage on DA reuptake efficiency. However, DA diffuses fast enough to escape from these dense regions after a few encounters, resulting in reduced DA reuptake.” We added a paragraph to the discussion to address this subject (line 447-452, line 463-468) and revised Supplemental Figure 1 and moved it as Figure 2 to the revised manuscript.

Both, the amplitude of the slow and the amplitude of the fast domain, directly depend on the amount of DA that was released in the synapses. The more DA is released, the higher is the amount of DA in the fast and the slow domain. In our data we identified the contribution of the slow domain in the continuously recorded FSCV data: it is the basis of the correlation between the wavelet power in the 0.5 Hz (2 seconds) frequency band and the 0.01 Hz (100 seconds) band.

Our statement: “net changes of extracellular DA concentrations occur at time scales of minutes to seconds and the amplitude of these changes is proportional to extracellular DA concentration...” is not at all trivial. The fact that extracellular DA concentrations change on time scales of minutes to seconds does not imply that the rate of removal (“amplitude of changes”) is proportional to the extracellular DA concentrations. This is only the case when DA is removed by a linear transport mechanism such as removal by DATs.

125-128. The model in Figure 1D is inaccurate and misleading, as is the associated text. There are no DA cells in the striatum – rather release is from presynaptic boutons and occasional synapses. Uptake of released DA is into DA axons, not cells. The inaccuracy of the premise underlying the model, call the validity of calculations from the model into question.

We thank the reviewer for identifying this inaccuracy. We incorrectly used the term “cell body” to label the intracellular space. However, this mislabelling does not affect any of the calculations. We revised the manuscript accordingly.

li. 151-163. The notion that the overall synaptic space is smaller than ECS volume is accurate. However, a more relevant reason for focusing on ECS for raclopride displacement is that the vast majority of D2 receptors are extrasynaptic, with less than 10% of DA D2 receptors associated with symmetric synapses (Yung et al 1995). This fundamental characteristic is not discussed, but should be. The authors then go on to say that DA is highest in concentration in a synapse, but this does not matter because the raclopride signal is only from the ECS. Given that DA receptors are predominantly extracellular, this makes an argument for the raclopride signal (reflecting displacement of binding by DA) as originating in the ECS. But instead, they argue that even though the highest DA concentrations, and necessarily the greatest raclopride displacement, will be in synapses, raclopride is primarily extrasynaptic so the signal will be from there. This is not convincing. Instead, there will be little synaptic contribution because 1) there are few postsynaptic specializations for DA release sites, and 2) DA receptors are not found there even when there are. Again, the conceptual underpinnings of the model call the model’s calculations into question.

The reviewer’s view on the distribution of D2 receptors between synapses and extrasynaptic extracellular space saying that the predominant fraction of D2 receptors is located extrasynaptically would fully support our theory. We state this already in the discussion (line 442-443): “It has been shown that the majority of D2Rs are located outside and often distant from synapses.” However, there is still a lot of controversy on this topic. In the paper of Yung et al. that the reviewer mentions the authors discuss the high extrasynaptic signal as potential staining artefact. Within the field of PET imaging the majority of researchers who use [11C]raclopride are still of the opinion that the signal reflects binding to intrasynaptic D2 receptors. Therefore we formulate our theory more carefully by saying that even if a large fraction of the D2 receptors would be located within synapses, the predominant part of the [11C]raclopride PET signal still originates from extrasynaptic binding because of the small synaptic volume.

We do not expect that the greatest [11C]raclopride displacement will be in synapses because (1) although DA concentration is highest within synapses the duration of high DA concentration (milliseconds) is too short to cause effective [11C]raclopride displacement and (2) the contribution of the intrasynaptic [11C]raclopride displacement to the PET signal would be low due to the small synaptic volume. We do not see conceptual deficits of the model here.

Figure 2. The data presented support activation of DAergic pathways with chemogenetic stimulation. However, some explanation for the rotational behavior should be given, as well as some comment on the greater enhancement of clockwise rotations, which seems odd as mouse dominant hemisphere is left as in most humans. An explanation is offered in the Supplemental Results, greater CNO-enhanced DA release in the left vs. right striatum. This should be noted when the data in Figure 2 are presented, along with an indication of why this might be the case e.g., (DAergic laterality, differential DREADD activation, etc.).

As was reviewed by Molochnikov and Cohen (2014), and supported by data from Glick and colleagues (Biochemical Pharmacology 23(1974) 3223-3225), animals show preferred rotational behaviour in the contralateral direction to the hemisphere with the highest DA content, this is also present with the addition of a DA system agonist (i.e. AMPH). As we show in revised Figure 9, CNO treatment causes significantly more DA release into the left striatum versus the right striatum (revised Supplemental Figure 3). This prevalence for left striatal DA release as induced by CNO treatment, correlates directly with the clockwise behaviour (contralateral to left side activation) that we see in these animals. It should be noted that 7/10 animals display this preference for clockwise rotation, fitting with previously published data in rats using AMPH treatment (7/12 in Gluck et al). Furthermore, in our PET studies, 4/5 animals which show an increase in rDA, show a higher increase in the left striatum, however one animal does display a slightly higher increase in the right striatum. These values also closely correspond to human literature, with right-handed dominance being linked to high DA in the left striatum. In the human studies, only right-handed individuals are selected for measurement, which would limit the amount of variability between hemispheres between subjects, selecting for individuals which likely have a left striatum dominance.

We expanded the Supplemental Results accordingly (lines 17-22) and referenced this discussion in the main manuscript (line 245-246).

255-272. An increase in frequency of DA transients with CNO is consistent with activation of DA neurons; but presumably the target is also expressed in DA axons, enhancing release. It is therefore surprising that an increase in amplitude was not seen. Might this reflect D2 autoreceptor activation for homeostatic regulation?

The reviewer is correct in postulating that the hM3D_{Gq} is likely also expressed in the axons of the dopaminergic neurons. However, there are four main points we would like to stress that differ our dataset from the few published reports using the hM3D_{Gq} in dopamine neurons. First, the approach we took in this study was to use low dose CNO (0.3 mg/kg), which is a much lower stimulus than the 1.0 mg/kg used in the only account of published dopamine specific, hM3D_{Gq}-expressing neurons (Calipari et al Nat Comms 2017). However, this low dose was sufficient to drive the locomotor response we would anticipate in these animals (Figure 3). Secondly, we are using heterozygous expression of the hM3D_{Gq} using a transgenic mouse model, whereas previous systems assessing CNO effects on the DA system used viral expression, which is known to cause a much higher expression level of the hM3D_{Gq} in the targeted region. Thirdly, we are detecting spontaneous and not electrically evoked transients. Lastly, but most importantly, we are assessing these effects using in vivo FSCV, where the only published report using the hM3D_{Gq} expression in dopamine neurons uses ex vivo slice recordings, which can also give slightly different results as compared to the in vivo physiology.

We cannot rule out that some aspect of D2 autoreceptor activation is occurring. However, the stimulus is robust enough to increase DA neuronal activity, by increasing the number of individual transients, but not to the same extent to cause a significant increase in the amplitude of individual signals.

303-304; 324-327. The frequencies examined were 0.5 and 0.01 Hz; it is therefore

unclear how either of these could be considered “high” frequency. “Higher” but not “high” might be appropriate.

To avoid ambiguities we introduce and clearly define the terms “high frequency” and “low frequency” in line 75-76 and also in line 328.

Figure 4. Why does the current envelope decrease with CNO? Also, how the purity of a DA signal was determined is not indicated; what is the evidence that DOPAC, an abundant DA metabolite did not contribute to the current transients detected?

The current envelope decreases independent of treatment and appears to be a consequence of the animal handling:

As described in the Methods, to isolate changes in dopamine concentration from other electrochemical signals, a principal component analysis was performed using a standard training set of stimulated dopamine release detected by chronically implanted electrodes, with dopamine treated as the first principal component among other unrelated electrochemical fluctuations such as changes in pH. Flow cell experiments using pure mixtures of dopamine and DOPAC have established that this approach can readily separate current changes attributable to dopamine from those of metabolites such as DOPAC (and HVA and 3-MT) and other neurotransmitters such as 5-HT (Heien et al., 2004).

Figure 7. Why does RDA/RDA baseline increase over time without CNO?

As discussed in line 424-430, noise contributes to rDA. And due to the radioactive decay of [11C]raclopride (half life = 20 minutes) rDA steadily increases over time. This effect is apparent in human and mouse data.

318-322. The text states that the highest increases in temporal variation in the raclopride PET imaging was in the left ventral striatum, but “highest” compared to what? Was this vs. right ventral striatum, or left or right dorsal striatum? Was dorsal striatum even monitored? If so, were changes significant? Was there laterality?

Temporal variations in the left ventral striatum were highest in the whole brain. Yes, there was laterality. In Supplemental Figure 3 we show the data of the right striatum.

339. The composition of the “tasteless” solution should be indicated in the text, particularly whether this was nutritive or not. This is also omitted from the methods section. As presented (346-353), the use of “tasteless” as the control solution implies that the difference is between “taste” and lack of taste. However, it is well known that insulin levels rise after ingestion of a nutritive solution, and that insulin crossed the blood brain barrier where it activates insulin receptor signaling and can amplify striatal DA release (e.g., Stouffer et al 2015; Woods et al 2016). This is likely to be a contributing factor to both the early and later phase increases in PET response. In this light, the most exciting aspect of the present studies is the differential timecourse of the PET response in ventral vs. dorsal striatum, although the explanation given omits the likely contribution of nutritive signaling.

The tasteless solution was non-nutritive and consisted of 25 mM potassium chloride and 2.5 mM sodium bicarbonate in 100%, 75%, 50%, or 25% dilution depending on the subjective feeling of tastelessness of each subject. We added this information to the Methods section (line 700) and referred to the paper by Tharanajah et al. for further details (line 711-712).

Minor

78. Should be “of” not “for”.

Changed accordingly.

121. No need to redefine ECS here.

We removed the repeating definition.

General. Figure should be referenced in the text individually in parentheses, rather as often presented here, at the end of a list of statistical results for a given experiment. It is difficult to find the corresponding figure that illustrates some sections.

We changed the references to Figure accordingly.

338. Human subjects were monitored, not measured.

We thank the reviewer for pointing out this inaccuracy and have updated the manuscript accordingly.

Reviewers' comments:

Reviewer #1 (Remarks to the Author):

The authors have satisfactorily addressed all the points I raised in my initial review.

Reviewer #2 (Remarks to the Author):

The manuscript by Backes and colleagues has been much improved by the addition of Figure 6 using electrical stimulation of dopamine release to bolster the prior data showing DREADDS-CNO induced elevations in dopamine. It is a shame that no further PET studies can be performed. The addition of references for human FSCV literature that were missing is noted. The manuscript has also been improved in clarity. I have no further critiques of the findings.

Reviewer #3 (Remarks to the Author):

The authors have performed an ambitious revision of their manuscript, with new data using electrical stimulation of the VTA providing additional confidence in the relationship between high and low frequency variation in DA levels. It would have been very informative if this approach had also been combined with PET - as of present this is still an unknown. However, I don't view this as necessary within the present publication.

Introduction, page 3, line 55-56: As commented on earlier, please note that the paradigm of measuring DA release using raclopride PET was initially designed for pharmacological challenges. Refs 7-11 do not describe "task-induced reductions". Also, I cannot see that any reference to the Yale method has been added to the introduction.

Regarding the discussion on correction for multiple comparisons: I agree that a post-hoc restriction of the research volume is artificial. That is the reason for why we should have pre-registered research plans. It is very difficult for me, or someone else, to know whether or not the analysis was adapted to the results, or if the present analysis is the one that was originally planned based on the literature.

I think the authors should still be more careful when drawing conclusions regarding the suitability of their method for measuring DA release in cortical regions. The authors have written "Several groups observed task-related changes of the absolute raclopride signal in extrastriatal regions" . In order to place this statement into context, please mention also the literature using high-affinity radioligands such as FLB, where there are important negative studies even when using the more powerful amphetamine challenge (Aalto et al, EJNMMI 2009).

A general comment is that much of the interpretation of the data is in the result section. Ideally, most of this should move to the discussion.

Reviewer #4 (Remarks to the Author):

The paper has been improved in this resubmission. Many of the reviewers' concerns have been addressed, the model is now more clearly explained, and new data have been added. Concerns about the overall view of DA transmission that undergird the conclusions remain, however.

1) The authors' perspective about the pattern of striatal DA release during normal DA neuron firing appears to be based on conclusions from local or pathway stimulation of DA axons, which results in a locally homogeneous increase in DA concentration, so that clearance is largely by uptake,

which can then be described by Michaelis-Menten kinetics. This is unlikely to be the case with local transients, for which clearance of DA from the area of a release site will include a large component of diffusion and consequent dilution of the release response. So slow uptake plays a role, but not necessarily a dominant one.

2) 117-122. I do not doubt the authors' conclusion that "the amplitude of low frequency variations of extracellular DA concentrations is directly proportional to (and is therefore a potential measure of) synaptic DA activity. " However, I question the conceptual underpinnings of their arguments. They say, "...depending on the local density of DATs, net changes of extracellular DA concentrations occur at time scales of minutes to seconds and the amplitude of these changes is proportional to extracellular DA concentration..." This circular sentence is not convincing. Moreover, there is no evidence that local increases in extracellular dopamine concentration from natural events last for minutes at physiologically relevant levels. Indeed, within seconds, the concentration remaining after a given release event would be vanishingly small from diffusion, dilution, and the DAT -- especially given that the influence of the DAT is greater on longer lasting transients that provide more time for the DAT more time to act. Consequently, the authors insistence that changes in dopamine concentration occur on a minute time scale (e.g., p. 16 and 17) seems more designed to justify the temporal resolution of PET than to describe a physiological process – even in regions outside of the striatum.

3) 447-452. The authors mention "hang-up" of DA concentration as part of their rebuttal, and have added this to the text. However, most investigators using FSCV consider this to be an electrode artifact issue, reflecting increased DA adsorption after a concentration increase. Support for explanation this comes from the return to baseline seen within seconds after stimulation when DA release is recorded using genetically encoded optical sensors (e.g., Patriarchi et al 2018; Sun et al 2018).

4) It should be noted that the DAT-cre mice used for chemogenetic stimulation have decrease DAT expression (Bäckman et al 2006) and function (O'Neill et al 2017), which would be expected to affect the dynamics of DA clearance.

5) The added electrical stimulation data from two mice are consistent with the authors' claims. However, n = 2 is unusually small for a paper submitted to a high profile journal, and the value weakened by the lack of corresponding PET data (despite the clear rationale for its absence).

We would like to thank the reviewers for their positive comments and their constructive criticism. We very much appreciate that taking into account the reviewer's suggestion helped us a lot to improve our manuscript.

Reviewers' comments:

Reviewer #1 (Remarks to the Author):

The authors have satisfactorily addressed all the points I raised in my initial review.

Reviewer #2 (Remarks to the Author):

The manuscript by Backes and colleagues has been much improved by the addition of Figure 6 using electrical stimulation of dopamine release to bolster the prior data showing DREADDS-CNO induced elevations in dopamine. It is a shame that no further PET studies can be performed. The addition of references for human FSCV literature that were missing is noted. The manuscript has also been improved in clarity. I have no further critiques of the findings.

Reviewer #3 (Remarks to the Author):

The authors have performed an ambitious revision of their manuscript, with new data using electrical stimulation of the VTA providing additional confidence in the relationship between high and low frequency variation in DA levels. It would have been very informative if this approach had also been combined with PET - as of present this is still an unknown. However, I don't view this as necessary within the present publication.

Introduction, page 3, line 55-56: As commented on earlier, please note that the paradigm of measuring DA release using raclopride PET was initially designed for pharmacological challenges. Refs 7-11 do not describe "task-induced reductions". Also, I cannot see that any reference to the Yale method has been added to the introduction.

We thank the reviewer for pointing out this inaccuracy and added "pharmacological" to the sentence in line 55-56. In the revised version that we submitted to the journal reference 16 in line 56 refers to the paper by Normandin, Schiffer, Morris where the Yale method was introduced although none of the authors was at Yale at this time. We now additionally added the Morris et al. 2013 reference.

Regarding the discussion on correction for multiple comparisons: I agree that a post-hoc restriction of the research volume is artificial. That is the reason for why we should have pre-registered research plans. It is very difficult for me, or someone else, to know whether or not the analysis was adapted to the results, or if the present analysis is the one that was originally planned based on the literature.

I think the authors should still be more careful when drawing conclusions regarding the suitability of their method for measuring DA release in cortical regions. The authors have written "Several groups observed task-related changes of the absolute raclopride signal in extrastriatal regions". In order to place this statement into context, please

mention also the literature using high-affinity radioligands such as FLB, where there are important negative studies even when using the more powerful amphetamine challenge (Aalto et al, EJNMMI 2009).

We added the references to negative extrastriatal results in response to amphetamine when high-affinity radioligands (FLB, Fallypride) were used (Aalto et al. and Slifstein et al., lines 483-485).

A general comment is that much of the interpretation of the data is in the result section. Ideally, most of this should move to the discussion.

Reviewer #4 (Remarks to the Author):

The paper has been improved in this resubmission. Many of the reviewers' concerns have been addressed, the model is now more clearly explained, and new data have been added. Concerns about the overall view of DA transmission that undergird the conclusions remain, however.

We thank the reviewer for the careful and considered criticism of our work, and for her/his appreciation for the revisions that were made. Since we are sure that many researchers in the field will share the reviewer's concerns, particularly with reference to the existence of a slow component in dopamine removal after phasic release, there is clearly a need to set out the conceptual reasoning in more detail not only to convince the reviewer but also the broad readership we hope will be interested in the study.

Accordingly, we have now added more evidence from literature that shows the existence of a slow component in cases where adsorption at the FSCV probe can be excluded as potential source. Furthermore, we substantiate our statement that heterogeneous spatial distribution of the dopamine transporter could be a potential source for the slow component by novel model calculations that take into account diffusion of dopamine after quantal release through extracellular space with heterogeneous distribution of dopamine transporters. These calculations show that (1) heterogeneous distribution of transporters is indeed capable of adding a slow component and that (2) the relative magnitude of the slow in relation to the fast component critically depends on the distance of the probe from the release site. This is in accordance with in vivo data where the location of the probe was modified between stimulations.

We hope that these data convince the reviewer of the potential existence of a slow component. However, we want to point out that in accordance with the reviewer we do not claim that the slow component is of physiological relevance.

Please find below our detailed point-by-point response.

1) The authors' perspective about the pattern of striatal DA release during normal DA neuron firing appears to be based on conclusions from local or pathway stimulation of DA axons, which results in a locally homogeneous increase in DA concentration, so that clearance is largely by uptake, which can then be described by Michaelis-Menten kinetics. This is unlikely to be the case with local transients, for which clearance of DA from the area of a release site will include a large component of diffusion and

consequent dilution of the release response. So slow uptake plays a role, but not necessarily a dominant one.

We completely agree with the reviewer that DA clearance depends on both diffusion and transport. Therefore equation 1 of the manuscript (as in Cragg and Rice, TINS, 2004) describes local DA kinetics depending on both diffusion and transport. This equation was used to calculate the plots of revised Figure 2. Driven by the steep gradients diffusion dominates the kinetics close to the release site while distant from the release site transport dominates the kinetics.

To substantiate this statement we now performed novel model calculations where we used the numerical solution instead of the analytical solution given by equation 1. This has the advantage that we can easily analyse the influence of spatial heterogeneity of the transport parameter where it is difficult or even impossible to derive an analytical solution. Equation 1 is only valid for spatial homogenous distribution of transporters. Results of these model calculations are presented in Supplemental Figure 4 (see below) of the revised manuscript.

First, we reproduced the analytical results with our numerical approach by calculating DA concentration at 20 μ m distance from the release site after a single quantal release of 9800 DA molecules, which successfully replicates the results of Cragg and Rice (TINS, 2004, Figure 1 a, right plot).

Cragg and Rice:

Our calculations (Supplemental Figure 4A):

The red and black traces are identical to the traces in our Supplemental Figure 4A. For their “Diffusion only” calculation Cragg and Rice applied an unspecific uptake constant of $k=0.007 \text{ s}^{-1}$. We additionally calculated the results for $k=0$ showing that these are effectively identical. Supplemental Figure 4B, which shows a time interval of 30 s after the release, indicates that in the absence of transport diffusion alone would always introduce a component which is removed at a minute-by-minute time scale. However, it also shows that

a homogenous high removal rate will completely remove this component. This supports our statement that DA transport and not diffusion plays the dominant role for the effective local DA clearance.

2) 117-122. I do not doubt the authors' conclusion that "the amplitude of low frequency variations of extracellular DA concentrations is directly proportional to (and is therefore a potential measure of) synaptic DA activity." However, I question the conceptual underpinnings of their arguments. They say, "...depending on the local density of DATs, net changes of extracellular DA concentrations occur at time scales of minutes to seconds and the amplitude of these changes is proportional to extracellular DA concentration..." This circular sentence is not convincing. Moreover, there is no evidence that local increases in extracellular dopamine concentration from natural events last for minutes at physiologically relevant levels. Indeed, within seconds, the concentration remaining after a given release event would be vanishingly small from diffusion, dilution, and the DAT -- especially given that the influence of the DAT is greater on longer lasting transients that provide more time for the DAT more time to act. Consequently, the authors insistence that changes in dopamine concentration occur on a minute time scale (e.g., p. 16 and 17) seems more designed to justify the temporal resolution of PET than to describe a physiological process – even in regions outside of the striatum.

Our calculations show that heterogeneous transport or heterogeneous expression of DATs leads to a decrease of the local DA clearance rate. In Supplemental Figure 4 (see below) of the revised manuscript we show the results for the case that the transporter density decreases exponentially with distance from the release site. In contrast to the case of high transporter density, a small fraction of DA is removed on a minute-by-minute time scale. Zooming-in into DAT staining images confirm heterogeneous distribution of DATs in all regions where DATs are expressed (e.g. Block et al., J Neurosci., 2015, Fig 3). (See also discussion in the revised manuscript lines 451-457).

We agree with the reviewer that the major fraction of the DA is cleared within seconds. We also agree with the reviewer that given the nearly negligible proportion in relation to the bulk DA, which is cleared within seconds, the physiological relevance of this small fraction is questionable and still highly debated in the field. Nevertheless, it is predicted from the calculations (Supplemental Figure 4, see below), it was detected in several experiments in the literature, and, due to the relatively slow kinetics of raclopride, it is sufficiently high to modulate the PET signal.

We are not at all claiming any physiological relevance of these minute time scale DA variations. But our calculations and our data (correlation of high and low frequency power) indicate that these minute time scale variations can be directly related to phasic release, and are definitely present regardless of the potential physiological relevance of the signal. And, as such, we can use it as a tool for detection of phasic changes in dopamine.

"...the amplitude of these changes is proportional to extracellular DA concentration": In fact, the amplitude of the changes ($dDA(x,t)/dt$) is proportional to the extracellular concentration ($DA(x,t)$) only if transport is the dominating process. This is not the case if diffusion dominates the dynamics. We added the terms in brackets to the revised manuscript (lines 120,121) to make clear that "amplitude of changes" means the temporal derivative.

3) 447-452. The authors mention “hang-up” of DA concentration as part of their rebuttal, and have added this to the text. However, most investigators using FSCV consider this to be an electrode artifact issue, reflecting increased DA adsorption after a concentration increase. Support for explanation this comes from the return to baseline seen within seconds after stimulation when DA release is recorded using genetically encoded optical sensors (e.g., Patriarchi et al 2018; Sun et al 2018).

Thank you for pointing this out. The reviewer is correct that at least part of the “hang-up” has been interpreted as adsorption and desorption effect of the electrodes. However, there are several indications that this effect cannot in isolation explain all of the observed minute scale variations.

(1) Bath et al. (Anal. Chem., 2000) show in vitro that indeed there was ad- and desorption when a step function of DA was applied. They corrected their in vivo data in two ways: either by repeating the measurement at high frequency repetition rate, which reduces the effect, or by applying the parameters determined in vitro to calculate the contribution of the effect and remove it from the data. In both cases there was still a residual minute time scale component left in the in vivo data as corrected dopamine levels were clearly above baseline at the end of the 10 seconds time interval they analysed.

(2) In Figure 2 in the manuscript we show recordings by Garris et al. in the MPFC and the striatum. The adsorption effect can only account for a fraction of the peak DA. In the MPFC the whole DA is cleared at a minute time scale. Adsorption cannot account for this.

(3) The data of Wu et al. (J. Neurosc. Meth., 2001) and of Robinson et al. (Clinical Chemistry, 2003) show that the minute scale contribution relative to the peak critically depends on the probe location. If the ‘hang up’ of dopamine were solely caused by adsorption it should scale with the peak height. However, there are locations at which the whole DA signal is cleared at a minute time scale.

Figure 3 of Robinson et al. Clinical Chemistry, 2003: all traces were recorded with the same probe at different positions. The slow component that is obvious in the 7.5, 7.65, and 7.8 traces cannot be caused by adsorption because adsorption requires a peak of DA concentration.

Our novel calculations show that the contribution of the minute time scale component of a measured transient critically depends on the distance from the release site. Further away from the release site exponential decrease of DAT density as function of distance from the release site leads to a more pronounced slow component relative to the peak, while close to the release site the contribution of the slow component appears negligible relative to the peak (Supplemental Figure 4C and D, see below). Thereby our model is in accordance with the observations by Robinson et al. (Clinical Chemistry, 2003). Our model replicates another aspect of the data presented in Figure 3 of Robinson et al.: when we compare the amplitude of the minute scale component recorded close to the release site with that recorded further distant, both are approximately identical (Supplemental Figure 4D, see below). The accordance of our model results with the measured data also support our statement that transport and not diffusion determine the minute time scale dynamics.

These aspects are part of the Discussion in the revised manuscript (lines 451-457 and lines 460-475).

We hope our arguments convincingly support the existence of a minute time scale clearance rate of DA after release. Obviously our arguments would be supported by data from a different technique with fast time resolution but which did not suffer from the complications of electrode absorption. In fact, while it must be interpreted with caution given the novelty of the technique, the data from optical sensors in Patriarchi et al. 2018 do appear to show a slow component (Figure 5D,E).

But again: we are not claiming that this minor fraction of the released DA has physiological relevance, per se, but it is an inevitable side effect of phasic release and thus can be used as a tool for detection of DA release events.

4) It should be noted that the DAT-cre mice used for chemogenetic stimulation have decrease DAT expression (Bäckman et al 2006) and function (O'Neill et al 2017), which would be expected to affect the dynamics of DA clearance.

Thanks for raising this point. In fact, the animals we use are from Dr. Nils-Göran Larsson (Ekstrand et al 2007) and not the Bäckman Dat-Cre which the reviewer has commented on. Although Ekstrand et al. showed a small decrease in midbrain DAT transcript levels, we do not see a decrease in DAT protein via Western blot. To our knowledge, publications using this specific model have not shown any Cre-specific effects and we have not noted any Cre-specific phenotypes in our research group. Importantly, we can be confident that the findings are not specific to this mouse line as we have demonstrated the application of our method in wild-type mice with electrical stimulation and also in human studies.

5) The added electrical stimulation data from two mice are consistent with the authors' claims. However, n = 2 is unusually small for a paper submitted to a high profile journal, and the value weakened by the lack of corresponding PET data (despite the clear rationale for its absence).

A sample size of 2 would be unusually small if our analyses relied on between-subjects measures. In fact, though, the effect was so robust that it was detectable even in individual animals. Nonetheless, to allay the reviewer's concern, we have now added data from an additional two mice, which increased the level of significance in the correlation between 0.5 Hz and 0.01 Hz wavelet power. This further supports our main message that phasic dopamine release induces minute-by-minute fluctuations in extracellular dopamine concentrations. (See revised Figure 6)

Supplemental Figure 4. Dopamine transients after quantal release in tissue with heterogeneous DAT expression

The model for quantal release of 9800 DA molecules is identical to that used by Cragg and Rice with the difference that the differential equation is solved numerically instead of using the analytical solution. In this way the model can be calculated for any spatial distribution of DATs. (A) Extracellular DA concentration during the first two seconds after quantal release at a distance of $20\ \mu\text{m}$ from the release site. The traces for high (black) and low (red) homogenous DAT expression are identical to those in Figure 1 of Cragg and Rice. Assuming exponential decrease of DAT expression as a function of distance from the release site reduces the peak height (blue). (B) Extracellular DA concentration during the first 30 seconds after quantal release. If DA kinetics is determined by diffusion only (dashed black), a noticeable fraction of DA is cleared at a minute time scale while at high homogenous DAT expression all DA is removed within 0.5 seconds (red). A small but noticeable minute time scale component is also present at exponential decrease of DAT expression from the release site (blue). C and D show extracellular DA concentrations $50\ \mu\text{m}$ from the release site. While the fast component, that determines the peak, critically

depends on the distance from the release site, the minute time scale slow component is nearly identical at 20 μm and 50 μm distance.

REVIEWERS' COMMENTS:

Reviewer #4 (Remarks to the Author):

The authors have thoughtfully addressed previous concerns with new (illustrated) data and additional experiments. Well done!